

# Diagnosing the Atlantic Meridional Overturning Circulation in density space is critical in warmer climates

Fernanda DI Alzira Oliveira Matos[1], Dmitry Sidorenko[1], Xiaoxu Shi[1,2], Lars Ackermann[1], Janini Pereira[3], Gerrit Lohmann[1,4], and Christian Stepanek[1]

[1]Alfred Wegener Institute, Helmholtz Centre for Polar and Marine Research, Bremerhaven, Germany
[2]Southern Marine Science and Engineering Guangdong Laboratory, Zhuhai, China
[3]Department of Earth and Environmental Physics, Physics Institute, Federal University of Bahia (UFBA), Brazil
[4]Department of Environmental Physics and MARUM, University of Bremen, Bremen, Germany

**Correspondence:** Fernanda DI Alzira Oliveira Matos (fernanda.matos@awi.de)

**Abstract.** The Atlantic Meridional Overturning Circulation (AMOC) plays a crucial role in shaping the global climate system by redistributing heat and influencing large-scale climate patterns. Using the AWI-CM3 model, we compare the AMOC strength under pre-industrial and *abrupt4xCO2* scenarios derived in depth (z-AMOC) and density ($\rho$-AMOC) space. Water mass transformations are assessed to analyze the impact of background climate on surface-induced and interior-mixing-induced transformations. We find that both location and strength of AMOC maximum are directly affected by the framework choice. AMOC weakening is seen under $4xCO_2$ forcing in both frameworks, but with only z-AMOC showing recovery, whereas $\rho$-AMOC oscillates around the minimum value that is reached after the spinup phase of 50 years. The variability of the z-AMOC maximum in the pre-industrial scenario correlates only with that at $26°N$ due to the flattening of isopycnals into constant depth levels in the subpolar North Atlantic. Diagnosing $\rho$-AMOC reveals a shutdown of interior mixing induced water transformations. This is absent in z-AMOC, which only shows weakened vertical fluxes. Despite different mechanisms, all timeseries are highly correlated under $4xCO_2$, indicating major density shifts. This is evident from weakening of downward diapycnal velocities at deep convection sites. This shutdown is attributed to the decline (increase) of heat (freshwater) flux contribution to surface-forced diapycnal water transformations, as a result of sea-ice melt and reduced poleward ocean heat transport to the subpolar North Atlantic. In contrast, the analysis of vertical velocities, used to derive z-AMOC, only reveals a general weakening of vertical fluxes, both upward and downward. Thus, z-AMOC and $\rho$-AMOC weakens, but driven by different mechanisms. Our results highlight the importance of diagnosing AMOC in density space, particularly in warmer climates. We recommend broader adoption of $\rho$-AMOC diagnostics across models and timescales to improve the understanding of AMOC response to climate change.

## 1 Introduction

The Atlantic Meridional Overturning Circulation (AMOC) plays a crucial role in the global climate, accounting for approximately 25% of global heat transport and nearly half of the deep-water formation in the ocean (Srokosz et al., 2012). The AMOC is also deemed as a global tipping point that is weakening and likely to collapse if subject to enough climate forcing,



even though the magnitude of weakening or potential of collapse is uncertain and under heavy discussion (McKay et al., 2022; Ditlevsen and Ditlevsen, 2023; van Westen et al., 2025; Dima et al., 2025; Zimmerman et al., 2025). Such potential weakening
under anthropogenic climate change has become a focus of increasing scientific attention as it is linked to catastrophic climatic events, particularly in economically vulnerable regions (Orihuela-Pinto et al., 2022; Meccia et al., 2025; Zhang et al., 2024). Understanding AMOC variability is therefore critical for assessing the potential environmental and societal impacts of future climate change (Bellomo et al., 2021).

At current background climate, the AMOC firstly consists of an upper limb of warm, saline waters flowing northward toward
the subpolar North Atlantic (Buckley and Marshall, 2016). This upper limb is fed by two primary pathways: the "cold route," which transports relatively cold and fresh waters from the Drake Passage in the Southern Ocean, and the "warm route," which carries relatively warm and saline waters from the Agulhas Plateau at the boundary between the Indian and Atlantic Oceans (Rühs et al., 2019). The maximum northward volume transport occurs within its mid-depth cell, at approximately 1000 meters below the ocean surface and between $30°$ to $65°$N (Matos et al., 2020). Upon reaching the northern North Atlantic, these
waters lose heat to the atmosphere, become denser, sink, and return southward as a cold lower limb that is dominated by the North Atlantic Deep Water (NADW) (Buckley and Marshall, 2016). Furthermore, meridional overturning in the Atlantic Ocean includes the abyssal cell that is fed by Antarctic Bottom Water (AABW) sourced from the Southern Ocean (Bilo et al., 2024).

The strength and variability of the AMOC are primarily driven by winds, buoyancy fluxes and interior mixing processes in the ocean that vary over synoptic to multi-centennial timescales (Megann et al., 2021). These are strongly influenced by
fluctuations in heat and freshwater fluxes along the Atlantic Ocean, which control the density of water masses transported by ocean currents (Sévellec and Fedorov, 2016). However, observing variations in buoyancy fluxes at fine resolution remains challenging due to current technological limitations, despite recent advances in AMOC fingerprint analysis and observational arrays (Frajka-Williams et al., 2019). As a result, numerical model simulations remain essential for studying the AMOC across multiple scales.

An increasingly critical factor to be considered in calculating the AMOC is the choice of vertical coordinate system used to derive AMOC-related output (Foukal and Chafik, 2024). The majority of AMOC estimates is provided in depth space, where the AMOC is calculated as a zonally averaged stream function varying with latitude and depth (z-AMOC hereafter) (Sidorenko et al., 2021). While this approach is straightforward, it may conceal the AMOC strength and variability, particularly at higher latitudes, because computing flow averages along inclined isopycnals in a depth-based coordinate system compromises the rep-
resentation of deep convection. Representing the AMOC in terms of density and latitude (density space; $\rho$-AMOC thereafter) provides therefore a more accurate depiction of global water mass transformations (Sidorenko et al., 2021; Foukal and Chafik, 2024). Moreover, deriving the overturning circulation in density space instead of depth space is more suitable not only for studies focused on the subpolar North Atlantic. Constant-depth averaging can also lead to spurious features such as the Deacon cell in the Southern Ocean (Döös and Webb, 1994; Stevens and Ivchenko, 1997; Speer et al., 2000) and has been linked to
discrepancies between modeled and observed AMOC variability across timescales (Liu et al., 2017).

Of particular relevance is a better understanding of the AMOC behavior under the current strong anthropogenic climate forcing. It is uncertain whether, and if so, when, significant weakening of AMOC will happen in a warming climate. One





question is, whether different interpretations of AMOC response are linked to the choice of analyzing ocean circulation either via z-AMOC or $\rho$-AMOC. Here, we address this question by investigating how the AMOC mean state and variability re-
spond to abrupt climate change when calculated in these two different frameworks. Our simulations were conducted using the AWI-CM3 model (Alfred Wegener Institute Climate Model version 3; Streffing et al. (2022)), following the Coupled Model Intercomparison Project version 6 (CMIP6) protocol for the *piControl* and *abrupt-4xCO₂* simulations (Eyring et al., 2016). Even though the severely perturbed warm climate state of *abrupt-4xCO₂* may seem drastic compared to the current climate trajectory, previous research has shown that it can be useful to contextualize ongoing climate change (Yang et al., 2023) and
evaluate the resilience of the AMOC to elevated $CO_2$ forcing while accounting for natural variability (Baker et al., 2025). Additionally, we aim at demonstrating the advantages and potential of adopting $\rho$-AMOC as standard output in intercomparison projects and observational networks (Foukal and Chafik, 2024; Frajka-Williams et al., 2023; Sidorenko et al., 2021).

## 2  Methodology

### 2.1  Description of Model and Simulations

Utilizing the model AWI-CM3 (Streffing et al., 2022), we conduct a quasi-equilibrium pre-industrial simulation (1850 CE; PI hereafter) that we employ as the control climate, and a perturbed warm climate simulation with an abrupt quadrupling of pre-industrial $CO_2$ concentration (*abrupt-4xCO2*, Eyring et al., 2016, *4xCO₂* hereafter). Both simulations span 200 years and were branched off from an equilibrated 1000-year PI spin-up simulation. The AWI-CM3 configuration employed here includes OpenIFS version 43r3v2 (Buizza et al., 2017) as the atmospheric component and FESOM2.5 (Danilov et al., 2017) as the
ocean and sea-ice components. Relevant fluxes are coupled between ocean and atmosphere via the OASIS3-MCT coupler (Craig et al., 2017).

OpenIFS was employed at approximately $100\,\mathrm{km}$ horizontal resolution on a cubic octahedral grid (TCo95L91) for higher computational efficiency (Malardel et al., 2016). FESOM2.5 was employed with an ocean mesh configuration of one-degree nominal resolution, refined near the Equator (1/3°), north of 50°N ($\sim 24\,\mathrm{km}$), and near coastlines. The ocean is discretized in
the surface in approximately 127,000 mesh nodes. Vertically, the model uses 46 depth levels implemented under an Arbitrary-Lagrangian-Eulerian (ALE) vertical coordinate scheme, which allows for flexible vertical layer configurations. For these simulations, we employed the z* vertical coordinate system (Adcroft and Campin, 2004), which scales vertical layers proportionally to sea surface height, thereby reducing spurious mixing in regions with strong stratification (Petersen et al., 2015).

To enable diagnostics of AMOC in density space, we remapped the model's vertical levels onto 89 density bins referenced to
2000 dbar ($\sigma^2$; $\mathrm{kg\,m^{-3}}$). These bins ranged from $30\,\mathrm{kg\,m^{-3}}$ to $40\,\mathrm{kg\,m^{-3}}$ and were constructed following methodologies described by Megann (2018), Xu et al. (2018), and Sidorenko et al. (2020a). The model outputs standard variables in depth space while computing transports in density space during runtime, thereby optimizing storage and computational costs (Sidorenko et al., 2021). In addition to the calculation of z-AMOC and $\rho$-AMOC, we also compute the contributions of surface-induced diapycnal ($\psi_S$) and interior-mixing-induced ($\psi_I$) water mass transformations that are related to the AMOC (hereafter referred



to as surface and interior transformations), following the mathematical and technical framework described in Sidorenko et al. (2020a).

## 2.2   Mathematical Framework

The $\rho$-AMOC, surface and interior transformations, and z-AMOC are calculated following the mathematical framework described by (Sidorenko et al., 2020a), using algorithms specifically designed for AMOC diagnostics on unstructured meshes

(Sidorenko et al., 2020b). For completeness, we provide a summary of the key definitions and equations used in this study:

$z$-AMOC represents the zonally-integrated meridional overturning streamfunction in depth space and is calculated using vertical velocity ($w$) according to Eq. 1 below:

$$\psi_z(y,z) = \int\limits_{North}^{South} \int\limits_{East}^{West} w(x',y',z)dx'dy' \tag{1}$$

Conversely, the $\rho$-AMOC represents the zonally-integrated meridional overturning streamfunction in density space and is

defined according to Eq. 2:

$$\psi_\sigma(y,\rho) = \int\limits_{North}^{South} \int\limits_{East}^{West} w_\rho(x',y',\rho)dx'dy' \tag{2}$$

where $w_\rho$ represents the diapycnal velocity across the density surface $\rho$.

Following the method by Sidorenko et al. (2020a), to calculate the interior transformations, ($\psi_I$), it is first necessary to determine the total diapycnal transport ($\psi_T$) and surface transformations ($\psi_S$).

The total transport is derived as the difference between $\psi_\sigma$ and the volume drift between the density classes $\frac{\Delta V}{\Delta t}$, as expressed in the following Eq. 3:

$$\psi_T(y,\rho) = \psi_\sigma(y,\rho) - \frac{\Delta V}{\Delta t} \tag{3}$$

The surface transformations, on the other hand, are calculated according to the following Eq. 4:

$$\psi_S(y,\rho) = \frac{1}{\Delta\rho} \int\limits_{East}^{West} \int\limits_{North}^{South} \int\limits_{\rho+\frac{\Delta\rho}{2}}^{\rho-\frac{\Delta\rho}{2}} F_p(x',y',\rho')d\rho'dy'dx' \tag{4}$$

where $F_p$ represents buoyancy flux, and $\Delta\rho$ is the size of the density bin. Finally, interior transformations are calculated according to the following Eq. 5:

$$\psi_I(y,\rho) = \psi_\sigma(y,\rho) - \frac{\Delta V}{\Delta t} - \psi_S(y,\rho) \tag{5}$$



## 3 AMOC in Density and Depth Space

The $\rho$-AMOC ($\psi_\sigma$) in the PI and 4xCO$_2$ simulations (Figures 1a and c) is characterized by a mid-depth clockwise cell lo-
cated within density intervals of $34\,\mathrm{kg\,m^{-3}}$ to $36.9\,\mathrm{kg\,m^{-3}}$ and latitudes of $40°$ to $60°$N, associated with the NADW (AMOC
maximum thereafter for simplicity). A secondary shallower maximum near $20°$N is also detected, representing the diapycnal
component of the subtropical gyre (Megann, 2018). The abyssal anticlockwise cell is centered around densities of $36.9\,\mathrm{kg\,m^{-3}}$
to $37\,\mathrm{kg\,m^{-3}}$ and extends up to approximately $60°$N. Another feature in Figure 1a is the Nordic Seas Overflow Waters
(NSOW), represented by the clockwise cell confined between latitudes of $60°$ to $80°$N and density range of $36.75\,\mathrm{kg\,m^{-3}}$
to $37.05\,\mathrm{kg\,m^{-3}}$. This feature arises due to topographic constraints in the region (Saunders, 2001).

Table 1 provides a detailed comparison of the location and strength of the $\rho$- and z-AMOC maximum (positive clockwise
streamlines) under PI and 4xCO$_2$ conditions. The $\rho$-AMOC maximum strength decreases significantly from $21.11\,\mathrm{Sv}$ in PI
to $7.66\,\mathrm{Sv}$ under 4xCO$_2$, accompanied by a shift in density from $36.68\,\mathrm{kg\,m^{-3}}$ to $35.87\,\mathrm{kg\,m^{-3}}$. Such upward shift of the
AMOC maximum can also be referred to as AMOC shoaling and is a common feature in studies with imposed radiative
forcing substantially higher than that of the pre-industrial climate (Matos et al., 2020; Baker et al., 2025). The latitude of the
maximum also shifts slightly northward, from $55°$N to $58°$N (Fig. 1a, c). This shift is associated with a northward expansion of
the upper cell, driven by sea ice loss and subsequent increase of deep convection in the Greenland-Iceland-Norwegian (GIN)
Seas (Chafik and Rossby, 2019).

**Table 1.** Location and strength of the $\rho$- and z-AMOC maxima, computed as the average of the last 50 years of each simulation.

|  | $\rho$-AMOC | |
| --- | --- | --- |
|  | Pre-Industrial | 4xCO$_2$ |
| Density (kgm$^{-3}$) | 36.68 | 35.87 |
| Strength (Sv) | 21.11 | 7.66 |
| Latitude (°) | 55 | 58 |
|  | z-AMOC | |
|  | Pre-Industrial | 4xCO$_2$ |
| Depth (m) | 910 | 790 |
| Strength (Sv) | 16.32 | 7.61 |
| Latitude (°) | 39 | 35 |

Similarly, the z-AMOC maximum strength declines from $16.32\,\mathrm{Sv}$ in PI to $7.61\,\mathrm{Sv}$ under 4xCO$_2$ conditions, with a corre-
sponding shoaling from $910\,\mathrm{m}$ to $790\,\mathrm{m}$ and with a southward shift in latitude from $39°$ to $35°$N (Fig. 1b, d). The mid-depth
cell is centered at depths of approximately $750\,\mathrm{m}$ to $1500\,\mathrm{m}$ and at latitudes of $30°$ to $45°$N, while the bottom cell is centered
near depths of approximately $4\,\mathrm{km}$, with a strength ranging from $\sim 2\,\mathrm{Sv}$ to $6\,\mathrm{Sv}$. The location of the z-AMOC maximum at a




**Figure 1.** Mean AMOC strength in units of Sverdrup ($1\,\mathrm{Sv} \equiv 1\times10^{6}\,\mathrm{m^3\,s^{-1}}$) averaged over the last 50 years of the PI and $4\mathrm{xCO_2}$ simulations. Panels show (a,c) $\rho$-AMOC and (b,d) z-AMOC for (a,b) PI and (c,d) $4\mathrm{xCO_2}$.





subtropical latitude in our simulation arises as a result of the averaging of the isopycnals in the subpolar North Atlantic. Even though a subtropical AMOC maximum is also featured in the z-AMOC representation of many ocean, climate, and Earth Sys-

tem models, it is not observed when AMOC is calculated in density space (Baker et al., 2025). Another way of visualizing not only the better latitudinal coherence of the $\rho$-AMOC maximum in PI but also its northward shift towards GIN seas and other before-mentioned consequences of increasing $CO_2$ forcing, is by remapping $\rho$-AMOC onto depth levels (Figure A1). This remapping is suggested to improve the understanding of the advantages of calculating $\rho$-AMOC, as it is more similar to the already established AMOC visualization plots (Foukal and Chafik, 2024) and we included it in the supplementary information

for convenience.

Additionally, Figures 1 a and c display a recirculation cell characterized by closed streamlines surrounding the AMOC maximum. However, there are notable differences between $\rho$-AMOC and z-AMOC in terms of their spatial distribution, owing to their distinct driving mechanisms. While the appearance of the recirculation cell in depth space arises due to vertical motion, in density space it is driven by diapycnal transformation (Xu et al., 2018). Therefore, the recirculation cell in $\rho$-AMOC is

located at higher latitudes, between $45°N$ and $60°N$, where diapycnal transformations towards denser waters occur (Figure 4). In contrast, the recirculation cell in z-AMOC is confined between latitudes of approximately $30°N$ and $45°N$. Our findings, both in depth and density space, corroborate those from other models with similar configurations (Xu et al., 2018; Sidorenko et al., 2020a, 2021; Megann, 2018; Megann et al., 2021; Baker et al., 2025).

Figures 1b and d highlight two major consequences of quadrupling atmospheric $CO_2$ concentrations to the AMOC in both

frameworks: the weakening and shoaling of the upper cell (e.g NADW), and the northward-upward expansion of the abyssal cell (e.g. AABW). To assess the consistency of these processes over time, we define two AMOC indices in both density and depth spaces: (1) $AMOC_{max}$, representing the maximum AMOC strength between $30°N$ and $65°N$, or subpolar AMOC; and (2) $AMOC_{26}$, representing the AMOC strength at $26°N$, or subtropical AMOC. The indices are then further adjusted in density and depth spaces as well in PI and 4x$CO_2$ to capture only the AMOC strength of the upper cell. Table 2 provides the upper and

lower depth and density limits used to derive both indices in our simulations.

Both indices were analyzed using their annual means and a 15-year running mean to assess multidecadal variability. A first-degree polynomial trend was fitted and subtracted from the multidecadal time series to investigate their correlation with each other using Pearson's correlation test. The amplitude of AMOC variability was further examined in terms of its standard deviation ($\sigma$, Figures 2a, b).

The $AMOC_{26}$ and $AMOC_{max}$ indices (Figure 2) reveal that the choice of framework significantly influences the interpretation of AMOC changes under different forcings. The $AMOC_{26}$ index timeseries (Figure 2a) exhibit interannual to multidecadal variability in both frameworks in PI, with an average $AMOC_{26}$ strength of approximately $13\,Sv$. However, in 4x$CO_2$, $\rho$-$AMOC_{26}$ weakens twice as much as z-$AMOC_{26}$ by the end of the simulation. Additionally, after the initial drop that manifests over the first $\sim 75$ model years of 4x$CO_2$, $\rho$-$AMOC_{26}$ strength oscillates steadily around a mean of $3\,Sv$, displaying

multi-decadal variability. In contrast, z-$AMOC_{26}$ shows only interannual variability in 4x$CO_2$ and even a partial recovery toward the end of the simulation, from about $5\,Sv$ in simulation year 75 to about $7\,Sv$ in simulation year 200. The similar $\rho$-$AMOC_{26}$ and z-$AMOC_{26}$ strengths during PI conditions are expected because isopycnals in subtropical regions are less



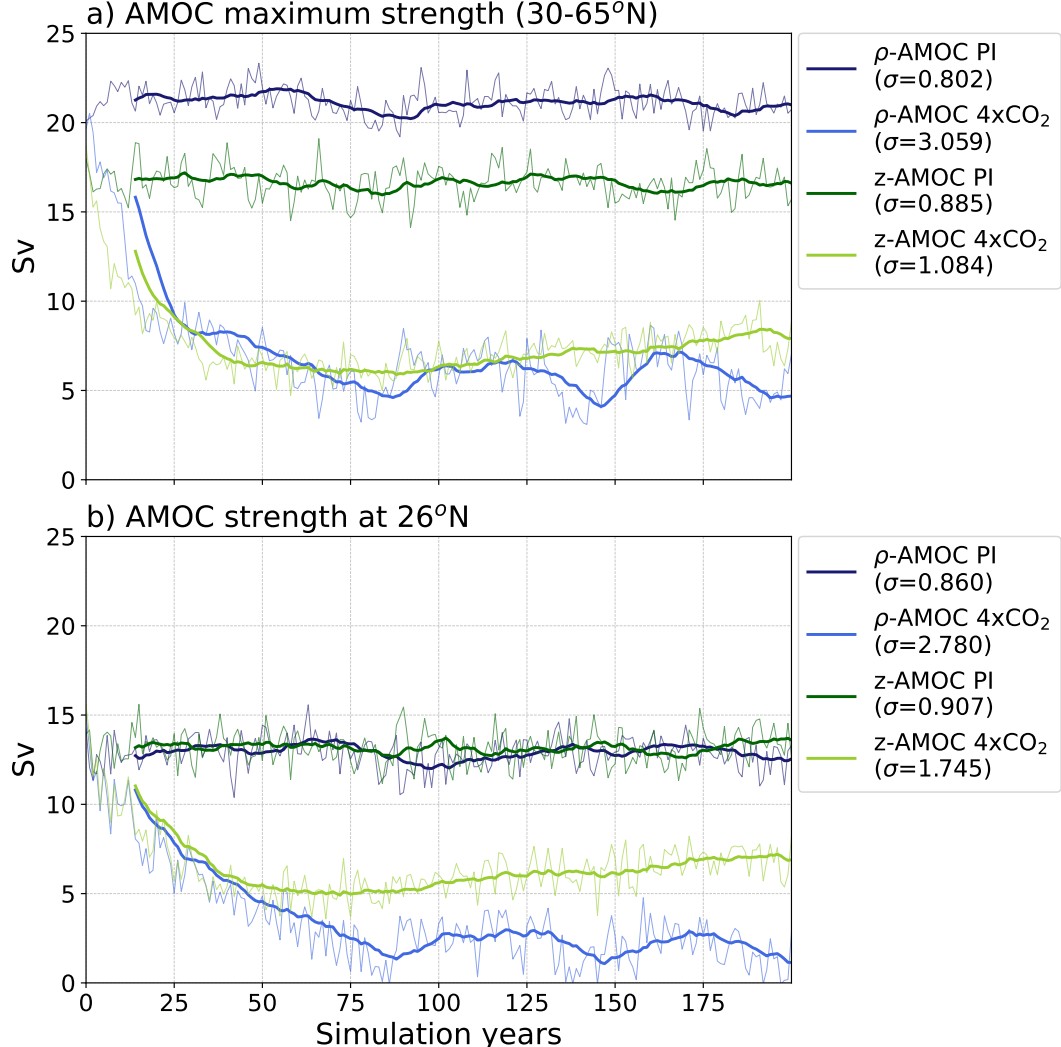

**Figure 2.** Annual mean time-series (thin lines) of $\rho$-AMOC (dark blue, royal blue) and z-AMOC (green, yellowgreen) in units of Sverdrup (Sv) under PI (dark blue, green) and $4xCO_2$ (royal blue, yellowgreen) forcing with superimposed 15-year rolling mean (thick lines). (a) AMOC strength at 26°N ($AMOC_{26}$ index), (b) maximum AMOC strength between 30° to 65°N ($AMOC_{max}$ index). Upper and lower depth and density limits for $AMOC_{26}$ and $AMOC_{max}$ in both simulations are shown in Table 2.

steep, or more parallel to z-coordinates, thereby increasing the equivalency of z- and $\rho$-AMOC (Wett et al., 2025; Moat et al., 2025). Furthermore, in subtropical regions, the upper and lower limbs of AMOC reside at different depth levels. This con-

trasts with subpolar regions, where northward and southward flows are horizontally separated (Lozier et al., 2019; Foukal and Chafik, 2024). Therefore, density changes in the water column have minimal impact on circulation patterns (Sidorenko et al., 2020a; Xu et al., 2018). However, because of their differing amplitude of variability, the timeseries of both $\rho$- and z-AMOC





at 26°N are not significantly correlated (Table A1). Alternatively, due to their low variability and the subtropical location of
the z-AMOC maximum, the timeseries of both AMOC indices in depth space exhibit the highest, even though still moderate,
correlation ($r = 0.57$, Table A1). The $\text{AMOC}_{max}$ timeseries (Figure 2b) shows that $\rho\text{-AMOC}_{max}$ is about $5\,\text{Sv}$ stronger than
z-$\text{AMOC}_{max}$ in PI conditions, with both displaying intra- to multidecadal variability. However, in $4x\text{CO}_2$, $\rho\text{-AMOC}_{max}$ and
z-$\text{AMOC}_{max}$ exhibit similar strength and are therefore more correlated ($r = 0.69$, Table A1) than in PI ($r = 0.28$, Table A1).
Despite this similarity, $\rho\text{-AMOC}_{max}$ maintains steady multi-decadal variability throughout the simulation period, as observed
with $\rho\text{-AMOC}_{26}$. On the other hand, z-$\text{AMOC}_{max}$ exhibits interannual variability and a gradual post-spinup recovery, also
similar to the evolution of z-AMOC26, which explains their high correlation ($r = 0.99$, Table A1).

**Table 2.** Upper and lower density and depth limits of the AMOC indices calculation, determined based the average of the last 50 years of
each simulation.

| | $\rho$-AMOC ($\text{kgm}^{-3}$) | |
| --- | --- | --- |
| | Pre-Industrial | $4x\text{CO}_2$ |
| Upper Limit | 36.00 | 35.50 |
| Lower Limit | 35.50 | 36.4 |
| | z-AMOC (m) | |
| | Pre-Industrial | $4x\text{CO}_2$ |
| Upper Limit | 500 | 500 |
| Lower Limit | 3000 | 3000 |

As figures 1 and 2 reveal different magnitudes of AMOC weakening and contrasting AMOC variability in $4x\text{CO}_2$ in depth
and density space compared to PI, the question of which framework is more physically meaning remains. When analysing the
influence of background climate on the AMOC, previous modelling studies have indicated that the extent of AMOC decline
under increased radiative forcing depends on initial AMOC strength and subsequent changes in large-scale climate patterns,
including surface air temperature, precipitation and sea-ice concentration (Bellomo et al., 2021; Zhao et al., 2024; Sigmond
et al., 2020). In these studies, models with stronger (weaker) AMOC in the control state tend to experience a greater (smaller)
degree of AMOC weakening. However, our modelling effort reveals that these conclusions might require revision once more
research institutes adopt the $\rho$-AMOC approach in their simulations. We certainly do not aim at questioning the discussions
arising from AMOC analyses in these or other studies, but simply refer to the fact that the AMOC is usually stronger in the
subpolar North Atlantic in density space in comparison to depth space, therefore, the degree of $\rho$-AMOC weakening tends to be
greater (Baker et al., 2025). Our PI $\rho\text{-AMOC}_{max}$ (stronger) and z-$\text{AMOC}_{max}$ (weaker) reach a comparable strength in $4x\text{CO}_2$
and since these two different AMOC magnitudes are found in the same simulation, the large-scale climatic consequences
are also the same. Therefore, these results indicate that calculating AMOC solely in depth space may underestimate both the
magnitude of AMOC weakening and its long-term impacts. In addition to the relation between AMOC initial strength and




subsequent weakening, Bellomo et al. (2021) demonstrated AMOC recovery occurring after surface air temperature stabilizes. However, even after reaching quasi-equilibrium, our simulations show that this recovery only holds true for z-AMOC, with $\rho$-AMOC showing damped oscillations around a nearly collapsed state, which contradicts the recent hypotheses of AMOC collapse in warmer climates (Ditlevsen and Ditlevsen, 2023). These damped oscillations are mainly attributed to feedbacks involving interactions between AMOC and Arctic and Antarctic sea-ice as well as the Atlantic Multidecadal Variability (AMV)

throughout the simulation period (Zhao et al., 2024; Nobre et al., 2023). Other climatic phenomena have been pointed as significant to drive AMOC weakening, especially in the common era, such as atmospheric blocking, meso- to submesoscale dynamics, and increase in atmospheric $CO_2$ driving sea-ice and ice sheet melting (Ionita et al., 2016; Dima et al., 2021; Gou et al., 2024; Ackermann et al., 2020). However, to derive such analyses, we must have model output at finer time and spatial resolutions, and transient simulations, respectively. Our simulations, however, were performed in a quasi-equilibrium

formulation without dynamical ice sheets, deriving annual $\rho$-AMOC output and at a relatively coarse resolution.

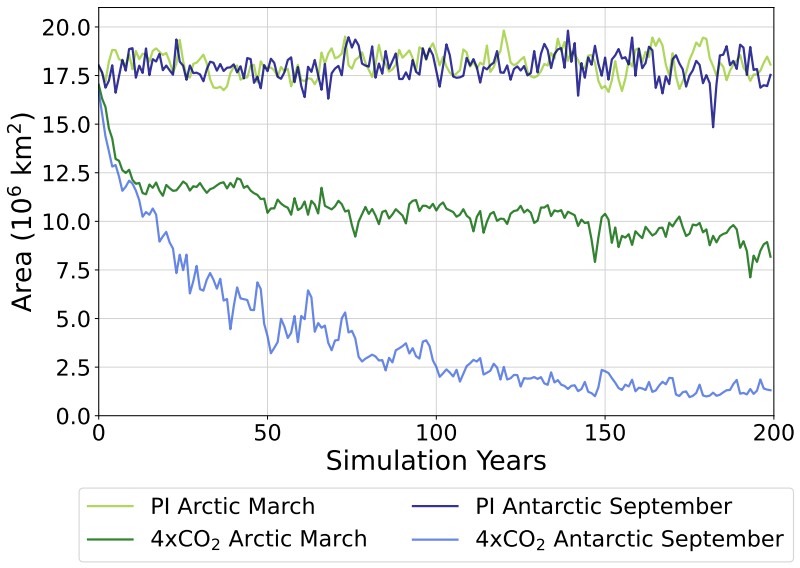

**Figure 3.** Winter Sea ice area in the Arctic and Southern Oceans in PI and $4xCO_2$ simulations.

In contrast to its spatial location, the maximum strength of the AMOC in $4xCO_2$ does not seem to differ between both frameworks (Table 1). While initially puzzling, this finding, when associated with sea ice loss in winter in both the Arctic and around Antarctica (Figure 3), indicates that AMOC in a $4xCO_2$ climate enters a new regime of persistently weak AMOC. Extreme sea-ice loss does not only increase the freshwater input into the Atlantic and Pacific via Arctic gateways but also

disrupts deep-water formation, a key mechanism for AMOC stability (Ionita et al., 2016; Mecking et al., 2017; Liu and Fedorov, 2022). Under future climate change scenarios, the AMOC is hypothesized to weaken due to increased freshwater forcing from melting of land and sea-ice and as a result of an intensifying hydrological cycle (Sévellec et al., 2017). However, its permanence around a weakened state is still unclear, with some models indicating an z-AMOC recovery due to the salinization of the surface



ocean layers driven by increased net-evaporation over the North Atlantic (Weijer et al., 2020; Ackermann et al., 2020). Even
though this AMOC recovery contradicts our $\rho$-AMOC results and supports our z-AMOC results, the magnitude and timescale
of this recovery vary greatly among the model simulations (Bonan et al., 2022; Zhao et al., 2024; Nobre et al., 2023), which
increases the uncertainty in the AMOC state in warm climates.

## 4    Large-scale and regional patterns of water mass transformations

As analysing the AMOC in density space directly accounts for water mass transformations, we are able to understand how
surface and interior transformations are thereby affected by abrupt climate change. The same analyses are not done in depth
space in this study firstly because interior mixing and surface buoyancy fluxes driven by heat and freshwater fluxes act primarily
on density surfaces, not depth levels, therefore, disentangling the two processes is more challenging in depth space (Walin,
1982). On the other hand, deriving interior mixing directly from the difference between total diapycnal transport and surface
transformations (Eq. 5) is a clearer and less convoluted process in density space.

Surface transformations ($\psi_S$; Equation 4; Figures 4a, c) reflect surface buoyancy fluxes and are confined to the surface mixed
layer. Consequently, they cannot be used to track water masses through the deeper ocean (Xu et al., 2018). They can, however,
indicate upwelling or downwelling of water masses and provide regional attribution through the maximum latitudinal extent of
each transformation cell. In the pre-industrial simulation (Figure 4a), three main surface transformation cells occur within the
limits of the AMOC upper limb: (1) a tropical cell centered at $31.5\,\mathrm{kg\,m^{-3}}$, (2) a subtropical cell centered at $35\,\mathrm{kg\,m^{-3}}$, and (3)
a subpolar cell centered at $36.94\,\mathrm{kg\,m^{-3}}$. The intermediate layer around $36.4\,\mathrm{kg\,m^{-3}}$ corresponds to Antarctic Intermediate
Water (AAIW), which forms in the Southern Ocean at intermediate density levels due to its low temperature and salinity
contrast (Santoso and England, 2004).

Surface transformations in PI (Figure 4a) occur toward lighter waters in the tropics, and toward denser waters in the sub-
tropical and subpolar regions. The same occur in the 4xCO$_2$ scenario (Figure 4c), albeit with a weaker and shallower subpolar
cell. In both simulations, heat fluxes dominate buoyancy-driven transformations over freshwater fluxes (Figure 5). In the trop-
ical cell, both heat and freshwater fluxes contribute to transformations toward lighter waters. This reflects the low salinity
(high net-precipitation) and high temperature (high net-heat flux into the ocean) that are typical of tropical regions with high
precipitation rates and significant river runoff. Conversely, within the subtropical cell, both fluxes act toward denser waters,
indicating a saltier, colder surface ocean. In subpolar regions, freshwater contributions are relatively small and are unable to
fully counteract heat flux-driven densification, setting the stage for deep mixing and deep water formation.



**Figure 4.** Mean surface ($\psi_S$; a, c) and interior ($\psi_I$; b, d) water mass transformation streamfunctions in (a, b) PI and (c, d) 4xCO₂, respectively. Blue contours indicate upwelling (i.e., transformations toward lighter waters), while red contours indicate downwelling (i.e., transformations toward denser waters).







**Figure 5.** PI and 4xCO2 (a,c) heat and (b,d) freshwater contributions to surface-forced diapycnal water mass transformations.

In the 4xCO$_2$ scenario, the large AMOC weakening reduces ocean meridional heat transport towards the subpolar North Atlantic until 65°N (Figure 6) increasing the heat flux contribution to surface transformations towards denser waters in the



subtropics (Figure 5b). As a result, surface transformations in the subpolar region are significantly reduced. Meanwhile, the contribution of freshwater fluxes to these transformations increases in the tropical and subpolar cells (Figure 5d), associated

with enhanced precipitation, southward shift of the ITCZ (Zhao et al., 2024), and sea-ice loss. Here, we only indicate sea-ice loss and increased hydrological cycle as key contributors to increased freshwater forcing in this region, as ice sheets are not interactive in our model, and therefore do not provide any additional freshwater flux due to ice volume loss.

Interior transformations ($\psi_I$; Equation 5; Figures 4b, d) account for water masses modified by interior mixing and cabbeling after being advected from regions influenced by surface transformations (Megann et al., 2021). In PI (Figure 4b), the interior

transformations peak near the $\rho$-AMOC maximum, indicating that the AMOC recirculation cell is driven mostly by interior mixing processes, while surface transformations are more dominant at higher densities. However, surface transformations also contribute to strengthening the AMOC by triggering mixing and subsequent deep convection. In subpolar regions, both surface and interior transformations drive deep-water formation, while subtropical regions exhibit interior transformations toward lighter waters, contributing to the shallower secondary maximum in this region. In tropical regions, surface transformations are

balanced by interior transformations, yielding less dense circulation (Figures 4a, b). At greater densities, interior transformations drive NSOW formation between $60°N$ and $80°N$, which subsequently feeds the lower NADW cell, while lighter water transformations are linked to AABW recirculation. Under the 4xCO$_2$ forcing (Figure 4d), surface transformations become more dominant, maintaining overturning, as interior mixing alone becomes insufficient to sustain deep convection (Figure 4c, d). While the subpolar cell remains driven by surface transformations, this is more strongly counterbalanced by interior mixing

in the subtropics, resulting in a weaker and shallower secondary maximum. Surface-forced transformations also contribute to AABW expansion, as interior-mixing-induced densification in the subpolar North Atlantic is significantly reduced.

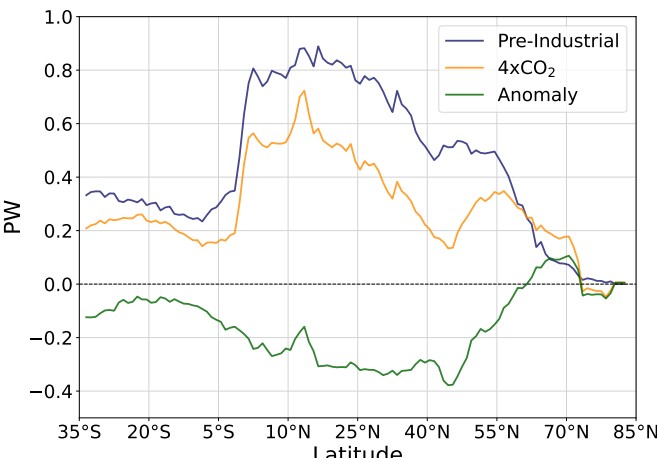

**Figure 6.** Atlantic Ocean meridional heat transport for PI and 4xCO$_2$.

As shown in Figure 4, surface transformations can trigger interior mixing at all density levels. Figure 7 provides a broader depiction of circulation dynamics in the North Atlantic, exhibiting $\psi_S$ (Figures 7a, b), diapycnal velocity ($w_p$; Figures 7c,





d), and vertical velocity ($w$; Figures 7e, f) at the density and depth levels of the maximum of the AMOC upper cell (Table

1). Please note that, in Figure 7, negative values (blue) indicate downward fluxes (buoyancy loss or downwelling), whereas positive values (red) indicate upward fluxes (buoyancy gain or upwelling).

In PI (Figure 7a), pronounced buoyancy loss occurs in the Labrador (LS) and Irminger Seas (IS), and along the Norwegian coast while weaker buoyancy gain is confined to areas near Greenland's east coast and parts of the North Atlantic Current (NAC). These transformations, occurring at the isopycnal surface of $\rho = 36.68\,\mathrm{kg\,m^{-3}}$, align with diapycnal velocities (Figure

7c), showing intensified downward fluxes due to interior mixing. Upward fluxes emerge in the Baffin Bay, along the Gulf Stream, NAC, and in the eastern GIN seas. The isopycnal surface of $\rho = 36.68\,\mathrm{kg\,m^{-3}}$ is shallower in the LS compared to the IS, consistent with observations (Lozier et al., 2019; Fu et al., 2023).

Vertical velocities at $\sim$910 m in PI (Figure 7e) exhibit much noisier patterns of upwelling and downwelling compared to diapycnal velocities (Figure 7c). However, a significant downward flux is evident south of Greenland, contributing to z-AMOC,

while upward fluxes dominate in LS and IS regions. Deep convection at these sites is still represented by a dipole pattern indicating upwelling (downwelling) in the southern (northern) LS and IS. Consequently, z-AMOC lacks a subpolar maximum, as opposing fluxes when integrated zonally result in weaker overturning strength. On the other hand, as the vertical motion from $30°$ to $50°$N is predominantly in the form of downwelling, the zonally integrated northward overturning is stronger in this region (Figure 1b), explaining the subtropical location of the z-AMOC maximum.

Under 4xCO$_2$ forcing, AMOC weakens similarly in both frameworks, but via different mechanisms. Diapycnal velocity (Figure 7d) shows reduced downward flux in subpolar regions due to weakened deep convection in the LS and buoyancy gain south of Greenland. Strong downward fluxes remain in the IS and GIN seas, primarily driven by surface transformations. Upward fluxes are also reduced along the path of the Gulf Stream. However, compared to PI, the vertical velocity patterns (Figure 7f), indicate reduced downward and upward fluxes across all regions, except the eastern North Atlantic. The reduced

downward flux near the AMOC upper cell maximum highlights shared mechanisms underlying AMOC weakening across both frameworks in 4xCO$_2$, albeit at different latitudinal bands. When evaluating the physical meaning of the patterns derived from diapycnal velocities against observations, we also see that this metric of vertical motion is able to capture the North Atlantic Ocean and GIN seas circulation changes under a warming climate. For instance, as suggested by Lozier et al. (2019), the Irminger Sea contributes more to deep convection than the Labrador Sea and this disparity is clear in our PI diapycnal velocity

field (Figure 7c). Our 4xCO$_2$ (Figure 7d), in turn, corroborates more recent studies stating that the IS predominant contribution to subpolar AMOC variability continues (Chafik et al., 2022; Sanchez-Franks et al., 2024) under increased radiative forcing. Additionally, as a consequence of sea-ice retreat and the subsequent albedo feedbacks and entrainment of water masses further North, the GIN seas deep convection is enhanced in warmer climates. Such change in GIN seas dynamics is not only visible in Figure 7d but also is suggested by the northward shift in the AMOC maximum in Figures 1 and A1. These findings are

then also corroborated by future projections of GIN seas overturning circulation (Årthun et al., 2023). Regarding the oceanic heat transport, with AMOC weakening, the Gulf Stream volume transport towards the subpolar North Atlantic also decreases (Piecuch and Beal, 2023), which is seen in Figure 7 via the reduced upwelling and downwelling in the subtropical North Atlantic, the weakened subpolar cell in Figure 1c, and the anomaly of the Atlantic oceanic heat transport in Figure 6. As





**Figure 7.** Mean surface-forced diapycnal water mass transformations ($\psi_S$; a, b), diapycnal velocity (c, d) and vertical velocity (e, f) under (a, c, e) PI and (b, d, f) 4xCO$_2$ conditions. Depth and density levels differ: PI (910 m, $\rho = 36.68\,\mathrm{kg\,m^{-3}}$) vs. 4xCO$_2$ (790 m, 35.87 kg m$^{-3}$). Negative values (blue) indicate downward fluxes (buoyancy loss), whereas positive values (red) indicate upward fluxes (buoyancy gain).



opposed to all the insights one can gain with the analyses of $\rho$-AMOC, z-AMOC fails to capture the before-mentioned dynamics
in our model and in previous studies (Sidorenko et al., 2020a, 2021). Therefore, our findings combined with evidence from
literature indicate that $\rho$-AMOC can provide a substantially more accurate representation of circulation changes, and AMOC
resilience and natural variability under warmer climates.

## 5   Conclusions

Here, we examined the patterns and variability of the AMOC and associated water mass transformations in depth (z-AMOC)
and density ($\rho$-AMOC) spaces using the AWI-CM3 model. Comparing these frameworks in different model configurations and
experiments is critical to evaluate the model skill in representing the AMOC under different boundary conditions and forcings.
Additionally, as the AMOC is fundamentally driven by diapycnal transformations, accounting for these transformations is
necessary to advance our understanding of mechanisms underlying AMOC strength regimes and variability across various
scales. Yet, most models employed at various resolutions and for past, present and future climate simulations only derive
z-AMOC as output, thus a community-wide adoption of $\rho$-AMOC is necessary to decrease model uncertainty in the future
(Foukal and Chafik, 2024). Conversely, observations have been increasing their efforts in calculating AMOC in density space,
even when the difference between $\rho$-AMOC and z-AMOC appears negligible, like at 26.5°N, where the RAPID array is
located (Moat et al., 2025; Frajka-Williams et al., 2023). Therefore, by continuing with z-AMOC only, the model-data discord
tends to increase, especially for measurements at higher latitudes. Such discrepancies raises important questions regarding the
reliability of our current understanding the AMOC, and our ability to project the response of large-scale ocean circulation
under the impact of climate change. To this extent, the *abrupt-4xCO2* experiment that we employed in this study aids in
understanding the sensitivity of the AMOC to abrupt climate change, providing valuable insights to its natural variability when
under adjustment to changes in sea-ice extent, freshwater fluxes onto the ocean, heat transport through ocean basins and global
teleconnections.

When comparing z- and $\rho$-AMOC, our results show significant differences between the two frameworks. In the pre-industrial
climate, the AMOC maximum occurs at different latitudes, with the z-AMOC ($\rho$-AMOC) maximum located in the subtropical
(subpolar) North Atlantic. Moreover, the AMOC is about $5\,\mathrm{Sv}$ stronger in density space. Under $4\mathrm{xCO_2}$ forcing, both frame-
works exhibit AMOC weakening and shoaling, and AABW northward-upward shift. However, z-AMOC variability aligns
more with subtropical patterns, whereas $\rho$-AMOC better captures the dynamics of the subpolar North Atlantic where isopy-
cnals are less parallel to depth coordinates. Since deep convection sites are located at higher latitudes, $\rho$-AMOC aids in the
deeper understanding of water mass transformations in the entire Atlantic whereas z-AMOC is more suitable to the subtropics.
Moreover, our results for $\rho$-AMOC align with previous studies using different models and resolutions, reinforcing the argu-
ment that computing AMOC in density space provides a more physically coherent depiction of the mechanisms driving AMOC
internal variability and stability (Xu et al., 2018; Roberts et al., 2020; Sidorenko et al., 2020a, 2021).

Even though a substantially warmer world might seem far-fetched, our findings in density space suggest that its effect on the
ocean circulation is not unrealistic and quite similar to the changes in overturning circulation occurring in the past decade and



projected to occur in the future. $\rho$-AMOC output indicates that under $4xCO_2$ forcing, the Greenland-Iceland-Norwegian Seas experience enhanced overturning that, together with the already critical contribution from the Irminger Sea, dominate deep convection in the subpolar North Atlantic. This northward shift in overturning is driven by sea-ice loss and subsequent reduced

winter sea-ice in the Arctic. The deep convection in the IS and GIN seas is maintained purely by surface transformations in $4xCO_2$, contrary to PI, where interior transformations drive deep convection aided by surface buoyancy fluxes, indicating enhanced stratification in the water column due to the added freshwater to the ocean with sea ice melt.

Despite these advantages, we acknowledge that $\rho$-AMOC computations require high frequency model output (Sidorenko et al., 2021). However, our findings highlight the advantage of adopting a framework that provides significantly more detailed

and physically meaningful information at similar computational costs compared to the depth space framework. The advantages are even more critical in warmer climates, when the climatic impacts under a relatively short timescale that accompany the current and rampant anthropogenically-induced radiative forcing, are better assessed and understood by including analyses of ocean circulation in density space rather than depth space. Indubitably, z-AMOC has provided meaningful advances in our understanding of the mechanism driving AMOC variability, and we do not advocate for its exclusion. Rather, we advocate for

including $\rho$-AMOC as model output in studies focusing on warmer climates, and observational diagnostics. A community-wide effort is crucial for our future, as both models and observations inform political, economic, and societal decision-making under climate change scenarios.

*Code and data availability.* Full documentation of AWI-CM3 is available at Streffing et al. (2022), except for the updates done in the ocean model FESOM2.5, available at https://github.com/FESOM/fesom2/releases/tag/AWI-CM3_v3.2 with a GPL-3.0 licence. All model

output, as well as the postprocessing and visualisation scripts, are archived on https://doi.org/10.5281/zenodo.15043857 (Matos, 2025a), https://doi.org/10.5281/zenodo.15044678 (Matos, 2025b), https://doi.org/10.5281/zenodo.15047024 (Matos, 2025c), and https://doi.org/10.5281/zenodo.15050831 (Matos, 2025d).



# Appendix A: Supplementary information

**Table A1.** Correlation coefficients for z- and $\rho$-AMOC timeseries. NS = not significant.

| | | Pre-Industrial | | |
| --- | --- | --- | --- | --- |
| | | $\rho$-AMOC | | z-AMOC |
| | | 26°N | Max | 26°N |
| z-AMOC | Max | NS | 0.275 | 0.568 |
| | 26°N | NS | 0.148 | |
| $\rho$-AMOC | Max | 0.378 | | |

| | | 4xCO$_2$ | | |
| --- | --- | --- | --- | --- |
| | | $\rho$-AMOC | | z-AMOC |
| | | 26°N | Max | 26°N |
| z-AMOC | Max | 0.619 | 0.693 | 0.986 |
| | 26°N | 0.642 | 0.715 | |
| $\rho$-AMOC | Max | 0.934 | | |





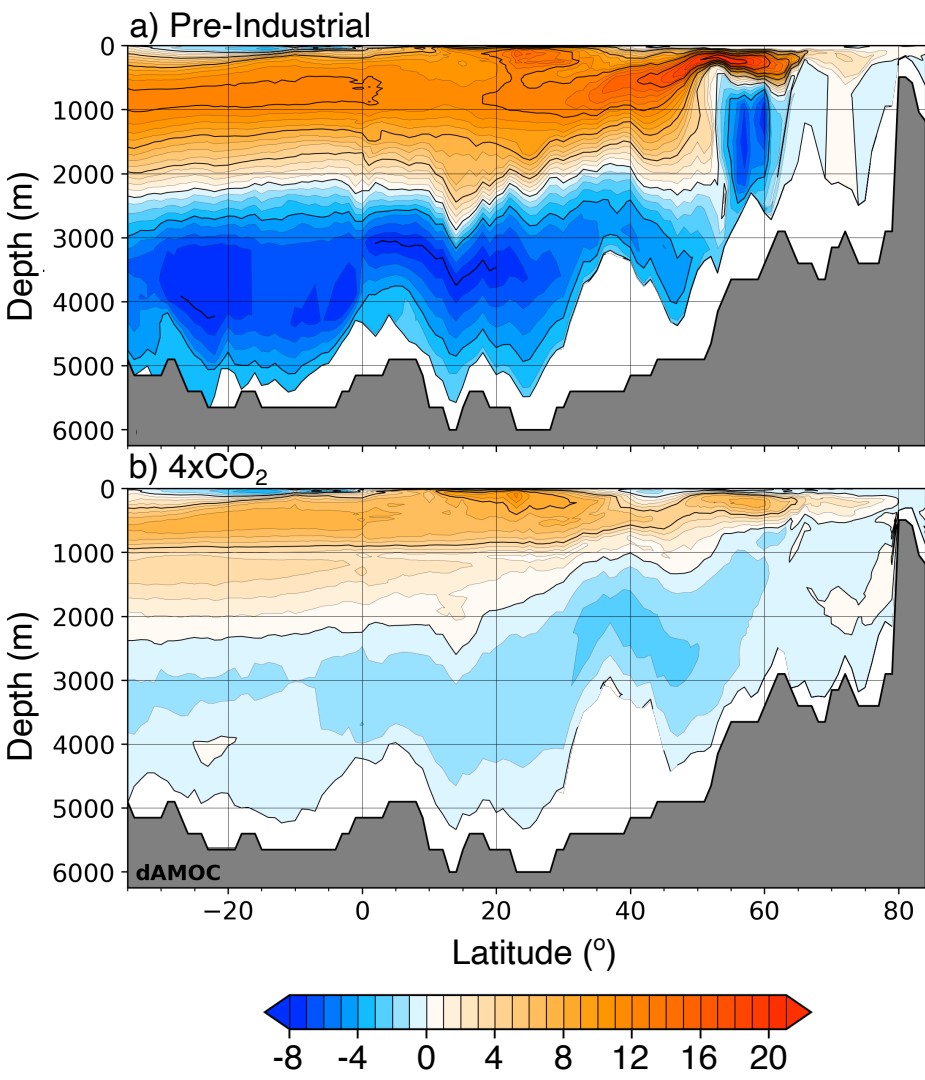

**Figure A1.** AMOC remapped onto depth levels for a) PI and b) 4xCO2 experiments.





*Author contributions.* Conceptualization and methodology, F.D.A.O.M., D.S.; Formal analysis, F.D.A.O.M., D.S., X.S., L.A.; Investigation,
F.D.A.O.M., D.S.; Resources, F.D.A.O.M., C.S., G.L.; Data curation, F.D.A.O.M.; Writing–original draft preparation, F.D.A.O.M.; writing–review and editing, F.D.A.O.M., D.S., X.S., L.A., J.P., G.L., C.S.; All authors have significantly contributed to the preparation of this manuscript.

*Competing interests.* The authors declare no conflict of interest.

*Acknowledgements.* The work was supported by the Helmholtz Association in its climate initiatives REKLIM (Regional Climate Change) (Q.
Wang, D. Sidorenko, C. Stepanek), and INSPIRES (International Science Program for Integrative Research in Earth Systems), at the Alfred Wegener Institute, Helmholtz Centre for Polar and Marine Research. The authors gratefully acknowledge the Jülich Supercomputing Centre (JSC) and the German Climate Computing Centre (DKRZ) for providing computing time on the supercomputers JUWELS and LEVANTE, respectively. The Python packages `pyfesom2` (https://github.com/FESOM/pyfesom2, last access: 04 February 2025) and `tripyview` (https://github.com/FESOM/tripyview, last access: 06 February 2025), as well as CDO version 2.4.1 (Schulzweida, 2023), last access: 04
February 2025) were utilized for post-processing and plotting of the data. We would also like to thank the Nippon Foundation and the Partnership for the Observation of the Global Ocean (POGO) for the support under the NF-POGO Centre of Excellence (NF-POGO CofE) programme.



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
