# Peer review of "Diagnosing the Atlantic Meridional Overturning Circulation in density space is critical in warmer climates"

_EGUsphere, 2025_

## Referee Comment (RC2)

**Summary**

This manuscript compares two different frameworks for diagnosing the Atlantic Meridional Overturning Circulation (AMOC): a latitude-depth streamfunction based on zonal integration and a latitude-density streamfunction based on density coordinates. They show that the structure of these circulations differ a bit in preindustrial circulations but highlight even larger differences in terms of the response of the circulation to climate change. They show that an additional advantage of the density-based MOC framework is that it relates directly to water mass transformations, which provide more mechanistic insights.

**Overview**

The analysis is interesting and technically correct. It will make a valuable contribution to the peer-reviewed literature. However, I think the contextualization of the results in terms of the broader literature needs to be improved before it can be accepted. Additionally, some of the results and implications of the work seem to be a bit overblown, and are less novel than the others make them out to be.

While I consider these concerns "major", they do not require any

**Major comments**

1) **Overselling how prominent z-AMOC diagnostics are.** The main motivation for this study is, as stated by the authors, that "The majority of AMOC estimates is provided in depth space". While they cite Sidorenko et al. (2021) here, that study does not actually provide any quantitative evidence in support of this claim. Instead, that study shows the difference between depth and density-AMOC in a single model. Their introduction cites a few papers that use depth-space analysis, but it is nowhere near the kind of exhaustive review you would need to make this statement. While the Foukal and Chafik (2024) paper is focused squarely on this question, they are also vague and qualitative. From where I sit in the field, z-AMOC is already well known to be a flawed diagnostic and anyone serious is already using rho-AMOC. Like Foukal and Chafik (2024), this paper concludes by advocating "for including rho-AMOC model output in studies focusing on warmer climates, and observational diagnostics". This does not recognize that the community is already doing this. OSNAP outputs their streamfunction in potential density coordinates and msftyrho is a CMOR variable that is already contributed to CMIP archives (although not as frequently as msftmz). I suggest the authors follow one of the two paths to address this, in addition to providing more context: either come up with a more rigorous estimate of how prominent z-AMOC is vs. rho-AMOV or else soften all of your language about how prominent z-AMOC is.

2) **The authors employ a different definition for rho-AMOC than most.** The conventional way of diagnosing rho-AMOC is by integrating meridional velocities (binned in density coordinates, as in msftyrho), not by integrating diapycnal velocities. Additionally, the authors do not explain how they diagnose their diapycnal velocities, which is non-trivial in models with a Lagrangian vertical coordinate. In fact, what they call the diapycnal velocity (following Sidorenko 2020) is different from what other people call the diapycnal velocity, because it is the Eulerian part of the diapycnal velocity that does not account for movement of isopycnal surfaces is time (Marshall 1999, Ferrari 2016). I recommend a clearer terminology and notation, perhaps reconciling yours with recent broader reviews on Water Mass Transformation methods (Groeskamp 2019, Drake 2025) that are not AMOC-specific. This is an important issue because the authors are advocating for more widespread adoption of these diagnostics but are advocating for different diagnostics than those used by most others.

3) **I am not convinced that the maximum rho-AMOC is a meaningful metric.** While the authors have indeed shown that the maximum z-AMOC and rho-AMOC are very different, a large fraction of this difference is due to the strong recirculation cell in rho-AMOC. This needs to be explained much more clearly. How should we think about what this means, conceptually or mechanistically? Is the formation part of this recirculation cell mixing via deep convection or via interior entrainment in overflows, for example? Why is this cell largely closed by diapycnal upwelling between 20ºN and 50ºN? Is this a region with strong interior mixing? If the point is to have a metric for the global-scale AMOC, wouldn't the transport that actually makes it out of the North Atlantic be a better metric of the circulation than something that largely reflects a local overturning cell?

4) **Vertical velocity is not a "mechanism", it is just the variable that feeds into the z-AMOC diagnostic.** I think referring to it as a mechanism actually weakens your argument. You should more forcefully emphasize that there is no mechanistic framework to quantitatively explain what causes the vertical velocities that feed the z-AMOC, whereas diapycnal transformations do provide a mechanism to understand the drivers of the rho-AMOC.

5) **The Figure with the rho-AMOC remapped into depth space should feature in the main text (e.g. as another column in Figure 1, although I would probably then swap the columns and rows).** Additionally, you should add a little more explanation of what this means in the caption. Presumably you compute the zonal-mean depth of each isopycnal at every latitude. This has become a very standard way of displaying the rho-AMOC and facilitates direct comparisons with the z-AMOC.

6) **Some of the discussion of the water mass transformations is more confusing that it is clarifying (see specific comments below).**

**Minor comments**

L. 33- "at approximately 1000 meters" is misleading, since that is where the streamfunction reaches its maximum, not the northward transport.

L. 153-155- What do you mean by "The indices are then further adjusted in density and depth spaces as well in PI and 4xCO2 to capture only the AMOC strength of the upper cell?" Shouldn't you have a generalizable metric that doesn't require manual adjustment in a different climate?

L. 168- Okay, but why doesn't this also apply to 4xCO2? Do isopycnals become more titled with climate change?

L. 258- What do you mean by "interior mixing alone becomes insufficient to sustain deep convection"? Why should we think about interior mixing sustaining deep convection in the first place? Isn't part of the deep mixing is the model *caused* by convection, i.e. unstable density profile triggers some kind of deep convection mixing scheme?

L. 262- What do you mean by "surface transformations *trigger* interior mixing"? Is this deep convection?

Figure 4- I think you need to expand on this either in the text or the caption to explain to readers how to read these plots, i.e. they are integrated from the North southwards. A meridional derivative in these quantities corresponds to diapycnal transformation whereas it being constant means there is no transformation.

L. 255-257 and Figure 4d- Are you saying that NADW is lighter than AABW? What is going on in the deep density layers? Is this Mediterranean overflow water that is mixing up at high latitudes? I don't really understand how to think about this.

L. 307- Be careful here, most of the energy that actually powers the AMOC circulation is mechanically input by Southern Ocean winds or interior turbulent mixing (Wunsch and Ferrari 2004).

It should be mentioned somewhere that what you call "interior mixing transformation" includes both parameterized physical mixing and spurious numerical mixing (and other residual errors).

Feel free to reach out to me for clarifications via email.

Henri Drake
UC Irvine
henrifdrake@gmail.com

---

## Author Comment (AC1)

**Author response for egusphere-2025-2326**

Fernanda DI Alzira Oliveira Matos, Dmitry Sidorenko, Xiaoxu Shi, Lars Ackermann,
Janini Pereira, Gerrit Lohmann, Christian Stepanek

July 1, 2025

**Letter to reviewer RC1**

Dear Reviewer,

We are extremely grateful to your suggestions at our manuscript upon submission to Geophysical Research Letters and now, to Ocean Science. Your in-depth comments have been invaluable in guiding us as to how to provide a more coherent description of the processes driving AMOC weakening under abrupt climate change and the importance of addressing it under density space. We hope that our response to this comment truly addresses all concerns raised and produces a substantially improved manuscript that advances our knowledge on this important subject.

Sincerely,
Authors

**Response to RC1's comments**

**1. The one suggestion of mine that the authors have not responded to is the one concerning the representation of convection in FESOM2.5. Section 2.1 needs to include a statement of how FESOM addresses density inversions: does it massively enhance vertical mixing, as, is done for instance, in NEMO? Or is there an explicit advective exchange of water parcels? I have been unable to find an explicit answer to this question in previous FESOM papers (e.g. Timmermann 2009, Sidorenko et al. 2014). In the context of the present paper, which distinguishes in detail between downwelling in depth space and density transformations in density space, I think such an explanation is essential.**

**#Response#:** In FESOM2.5, the K-profile parameterization (KPP; Large et al., 1994) scheme was employed for vertical mixing. Consequently, convection is parameterized through enhanced vertical mixing rather than explicit advection of water parcels. In practice, when convective conditions arise, the model artificially increases the vertical diffusivity (on the order of $0.01\,\mathrm{m^2\,s^{-1}}$) to homogenize the water column. This approach is common in hydrostatic ocean models, which cannot explicitly resolve convective plumes or advective exchange across isopycnals due to the hydrostatic approximation. Fully resolving convective

overturning would require a non-hydrostatic model with horizontal grid spacing comparable to the vertical scale ($\sim 1\,\mathrm{m}$), an unattainable resolution for global simulations on current supercomputers.

We added the following text into Section 2.1:

*"In FESOM, the K-Profile Parameterization (KPP; Large et al., 1994) scheme is employed for vertical mixing. Consequently, convection arising from local static instability is parameterized through locally enhanced vertical diffusivity, set to $0.01\,\mathrm{m^2\,s^{-1}}$."*

**2. Technical corrections**

**L304. "Here, we examined..." is awkward. I suggest replacing with "We have examined...".**

**L309. "Yet..." would read better if it were replaced by "Nevertheless...".**

**L342. "and rampant" is unnecessary – I would suggest deleting it.**

**Response#: We appreciate the corrections and have replaced the terms accordingly.**

**References**

Large, W. G., McWilliams, J. C., and Doney, S. C.: Oceanic vertical mixing: A review and a model with a nonlocal boundary layer parameterization, Reviews of Geophysics, 32, 363–403, https://doi.org/https://doi.org/10.1029/94RG01872, 1994.

---

## Author Comment (AC2)

**Author response for egusphere-2025-2326-RC2**

Fernanda DI Alzira Oliveira Matos, Dmitry Sidorenko, Xiaoxu Shi, Lars Ackermann,
Janini Pereira, Gerrit Lohmann, Christian Stepanek

August 10, 2025

**Letter to RC2**

Dear Reviewer,

We deeply appreciate your comments and suggestions to our manuscript that greatly helped us to improve the level of detail required for the understanding of our methodology and results. Please find below our responses to your comments.

Sincerely,

Fernanda D. A. O. Matos

(On behalf of the authors)

**Response to RC2's major comments**

*1. Overselling how prominent z-AMOC diagnostics are.* The main motivation for this study is, as stated by the authors, that "The majority of AMOC estimates is provided in depth space". While they cite Sidorenko et al. (2021) here, that study does not actually provide any quantitative evidence in support of this claim. Instead, that study shows the difference between depth and density-AMOC in a single model. Their introduction cites a few papers that use depth-space analysis, but it is nowhere near the kind of exhaustive review you would need to make this statement. While the Foukal and Chafik (2024) paper is focused squarely on this question, they are also vague and qualitative. From where I sit in the field, z-AMOC is already well known to be a flawed diagnostic and anyone serious is already using $\rho$-AMOC. Like Foukal and Chafik (2024), this paper concludes by advocating "for including $\rho$-AMOC model output in studies focusing on warmer climates, and observational diagnostics". This does not recognize that the community is already doing this. OSNAP outputs their streamfunction in potential density coordinates and msftyrho is

a CMOR variable that is already contributed to CMIP archives (although not as frequently as msftmz). I suggest the authors follow one of the two paths to address this, in addition to providing more context: either come up with a more rigorous estimate of how prominent z-AMOC is vs. rho-AMOV or else soften all of your language about how prominent z-AMOC is.

**#Response#:** We acknowledge the validity of the reviewer's observation and concede that the original phrasing regarding the community's reliance on z-AMOC was overstated and too general. Our study primarily focuses on comparisons between our findings and AOGCM/CMIP models, where the depth-coordinate streamfunction (e.g., msftyz/msftmz), as you observed, remains the standard and most frequently utilized AMOC diagnostic. While the density-coordinate streamfunction (msftyrho/msftmrho) was defined in the CMIP6 request (Griffies et al., 2016), it is, as you observed, archived less consistently and its presence is model- and experiment-dependent. This prevents the usage of $\rho-$AMOC output as extensively for research and model-intercomparisons as is possible for the z-AMOC output. For example, in Baker et al. (2025), only 5 of the 35 models featured in the article had $\rho$-AMOC output for the *abrupt-4xCO2* experiment. More generally, we also find evidence for less wide-spread production and usage of $\rho$-AMOC vs. z-AMOC from CMIP6-archive download statistics. For example, while Omon.msftmz is available for 39 models and has download rank 240[1] (i.e., 239 other variables were downloaded more often than Omon.msftmz), Omon.msftmrho is available only for 8 models and has been downloaded much less often (download rank 1159[2]). We do not believe that this disparity reflects a lack of community interest, and is likely attributable to the additional computational cost associated with deriving $\rho$-AMOC output. Consequently, a complete transition to $\rho$-AMOC in model intercomparisons and in research remains an ongoing challenge, especially within large model ensembles and for studies that require long integration periods and/or high resolution. We believe that the mentioned computational cost explains, at least partially, the continued use of z-AMOC in AOGCM studies, even as many researchers prefer $\rho$-AMOC when feasible, and when high-frequency output can be better resolved, as suggested in Megann (2024). Furthermore, based on our experience, the representation of AMOC in depth space remains widely applied even when models provide output in density space, owing to the community's familiarity with this representation.

While we believe that there is some merit in repeatedly highlighting the continued use of z-AMOC in many fields of climate research, we have refined and softened our language throughout the revised manuscript to emphasize that our statements regarding the application of z-AMOC stem mostly from coupled-model-based assessments from the CMIP and related model-intercomparison efforts. We now also recognize in our manuscript the strong momentum towards wider acceptance of the density-space based diagnostics in recent work, highlighting as examples the OSNAP and RAPID arrays (Frajka-Williams et al., 2019), and several modeling studies, such as Jackson and Petit (2023), Fu et al. (2023), van Westen et al. (2025). Such acknowledgment, along with a reference to Griffies et al. (2016) for a technical perspective on the CMIP6 data request, has been incorporated into the introduction and discussion sections.
* * *
[1] https://airtable.com/appOcSa4gXyzHThmm/shrkayKObes58Zu45/tblpo5L8maBIGlM1B/viwNNzrqK5oPL7zk2/recJ5HHT1QhjvpmvJ, accessed Aug 7, 2025

[2] https://airtable.com/appOcSa4gXyzHThmm/shrkayKObes58Zu45/tblpo5L8maBIGlM1B/viwNNzrqK5oPL7zk2/recJO0yp8BBdOE7sB, accessed Aug 7, 2025

***2. The authors employ a different definition for $\rho$-AMOC than most.*** **The conventional way of diagnosing $\rho$-AMOC is by integrating meridional velocities (binned in density coordinates, as in msftyrho), not by integrating diapycnal velocities. Additionally, the authors do not explain how they diagnose their diapycnal velocities, which is non-trivial in models with a Lagrangian vertical coordinate. In fact, what they call the diapycnal velocity (following Sidorenko 2020) is different from what other people call the diapycnal velocity, because it is the Eulerian part of the diapycnal velocity that does not account for movement of isopycnal surfaces is time (Marshall 1999, Ferrari 2016). I recommend a clearer terminology and notation, perhaps reconciling yours with recent broader reviews on Water Mass Transformation methods (Groeskamp 2019, Drake 2025) that are not AMOC-specific. This is an important issue because the authors are advocating for more widespread adoption of these diagnostics but are advocating for different diagnostics than those used by most others.**

**#Response#:** We appreciate the reviewer's insightful comment, and want to emphasize the robust equivalence of diagnosing MOC using vertical velocities in comparison to the more conventional usage of meridional (horizontal) velocities. Therefore, we expand below on three main aspects:

1. **Equivalence of the two approaches.** Using vertical velocities to compute MOC offers a computationally sound and completely equivalent alternative to integrating binned horizontal velocities. Such equivalence is demonstrated in Sidorenko et al. (2020a,b), wherein the authors explained that binning of horizontal divergence into density classes is done using instantaneous isopycnals, enabling diapycnal velocity calculation after removing the mean drift of isopycnals. The latter is negligibly small in our simulations. Because binning is done with respect to instantaneous isopycnals our AMOC diagnostics are equivalent to those derived using the horizontal velocities. We furthermore note that the use of vertical velocities, instead of horizontal velocities, is more a necessity than a deliberate choice due to the structure of the FESOM2 (see below).

2. **Vertical coordinate in FESOM2.** We apologize for any confusion: FESOM2 does not use Lagrangian vertical coordinate. We implement the Arbitrary Lagrangian-Eulerian (ALE) scheme in a finite volume sense (see lines 80-82 of the original manuscript and Scholz et al. (2019) for more information).

3. **Concern using horizontal velocities.** While computing $\rho$-AMOC using horizontal velocities is feasible and in principal equivalent to MOC computation with vertical velocities, this approach on an unstructured grid, such as the FESOM2 mesh we employ requires careful and non-trivial "broken-line" integration along control-volume boundaries (Sidorenko et al., 2020b) following the discretization of the continuity equation. Furthermore, doing so is less advantageous as it conceals critical information concerning diapycnal velocities. Additionally, diagnosing MOC in density space using vertical velocities has proven more efficient for online diagnostics on the FESOM2 unstructured mesh (Sidorenko et al., 2020a).

We hope this clarifies both the theoretical equivalence and the practical motivations for our chosen diagnostic. We did not, however, include this discussion in our manuscript, for brevity and because it is not within the scope of our study. In the revised manuscript, we included the equation to diagnose MOC using horizontal velocities in the appendix and include the following text to section 2.2:

"In this study, the AMOC is diagnosed using vertical velocities, as opposed to the conventional application of meridional velocities (equations using meridional velocities are detailed in the Appendix). This methodology, while mathematically equivalent, presents notable benefits, especially concerning the unstructured FESOM2 mesh, as it directly incorporates diapycnal velocities, and facilitates more efficient online diagnostics (Sidorenko et al., 2020a)."

**3. I am not convinced that the maximum ρ-AMOC is a meaningful metric.** While the authors have indeed shown that the maximum z-AMOC and ρ-AMOC are very different, a large fraction of this difference is due to the strong recirculation cell in ρ-AMOC. This needs to be explained much more clearly. How should we think about what this means, conceptually or mechanistically? Is the formation part of this recirculation cell mixing via deep convection or via interior entrainment in overflows, for example? Why is this cell largely closed by diapycnal upwelling between 20ºN and 50ºN? Is this a region with strong interior mixing? If the point is to have a metric for the global-scale AMOC, wouldn't the transport that actually makes it out of the North Atlantic be a better metric of the circulation than something that largely reflects a local overturning cell?

**Response#: We acknowledge the reviewer's concern regarding the distinction between the $\text{AMOC}_{max}$ index and the net export across the North Atlantic basin. Nevertheless, we argue that in our manuscript, the index is a valuable metric for emphasizing the importance of the AMOC in density space, as opposed to depth space. Perhaps, the definitions of surface-forced diapycnal and interior-mixing-induced water mass transformations, as well as interior mixing, may have caused confusion.**

Figure 4 of the manuscript presents the mean surface and interior transformations, which, when summed together with the model drift (that is deemed negligible and therefore not considered in the manuscript), yield manuscript Figure 1, panels a and b. Specifically, for pre-industrial (PI) conditions, between $20 - 50°\text{N}$, diapycnal upwelling is confined to density classes ranging from $33\,\text{kg m}^{-3}$ to $36\,\text{kg m}^{-3}$, where water mass transformations are predominantly surface-forced. Consequently, this recirculation cell is primarily driven by surface buoyancy fluxes confined to the surface mixed layer depth, rather than interior entrainment of overflows, which are incorporated within the interior-mixing-induced water transformation component of the AMOC in density space. For clarification:

- Surface transformations, denoted as $\psi_S$ in Figure 4, represent the component of the AMOC in density space that is forced by buoyancy fluxes confined to the surface mixed layer depth.

- Interior transformations, denoted as $\psi_I$ in Figure 4, represent the component of the AMOC in density space that is forced by mixing with water masses that are advected from other regions or modified through thermobaricity and cabbeling.

Both surface and interior transformations can contribute to the formation of denser waters (positive recirculation cell represented in red) and to the formation of lighter waters (negative recirculation cell represented in blue). Given that the aforementioned recirculation cell is confined to a lighter density class than the maximum ρ-AMOC as defined by the index, it does not affect the metric, and consequently, this subtropical cell (associated with the Subtropical Mode Water formation) is not extensively addressed in the manuscript.

In our PI simulation, the maximum ρ-AMOC, represented by the recirculation cell confined between $40 - 60°\text{N}$ and $36.20\,\text{kg m}^{-3}$ to $36.90\,\text{kg m}^{-3}$, is formed partially by surface buoyancy fluxes, but is primarily caused by interior transformations (including interior entrainment of overflows), as clear through comparing

figures 4a and b. Under $4xCO_2$ conditions, interior transformations act towards lighter waters, thereby weakening the positive recirculation cell in Figure 1. Overall, Figure 1a represents the sum of Figures 4a and 4b, while Figure 1b represents the sum of Figures 4c and 4d. As the changes in the $\rho$-$AMOC_{max}$ index capture this significant shift in the driving mechanisms of AMOC strength (surface versus interior transformations), we maintain that it is a meaningful metric for the global-scale AMOC.

*4. Vertical velocity is not a "mechanism", it is just the variable that feeds into the z-AMOC diagnostic.* **I think referring to it as a mechanism actually weakens your argument. You should more forcefully emphasize that there is no mechanistic framework to quantitatively explain what causes the vertical velocities that feed the z-AMOC, whereas diapycnal transformations do provide a mechanism to understand the drivers of the $\rho$-AMOC.**

**#Response#:** We appreciate the reviewer's feedback. To address this issue we modified the sentences where we refer to vertical velocity as a mechanism.

*5. The Figure with the $\rho$-AMOC remapped into depth space should feature in the main text (e.g. as another column in Figure 1, although I would probably then swap the columns and rows).* **Additionally, you should add a little more explanation of what this means in the caption. Presumably you compute the zonal-mean depth of each isopycnal at every latitude. This has become a very standard way of displaying the $\rho$-AMOC and facilitates direct comparisons with the z-AMOC.**

**#Response#:** We thank the reviewer for raising this point. We have updated Figure 1 to include the $\rho$-AMOC remapped into depth space and updated the figure caption:

[Figure]

Figure 1: Mean AMOC strength in units of Sverdrup ($1\,\mathrm{Sv} \equiv 1 \times 10^6\,\mathrm{m}^3\,\mathrm{s}^{-1}$) averaged over the last 50 years of the PI and 4xCO$_2$ simulations. Panels show (a, b) $\rho$-AMOC, (c,d) $\rho$-AMOC remapped into depth coordinates, and (e, f) z-AMOC for (a, c, e) PI and (b, d, e) 4xCO$_2$. In (c,d), $\rho$-AMOC is remapped into depth coordinates by loading the mean layer thickness of each density class, cumulatively summing these thicknesses to obtain the bottom depth of each class, then shifting the cumulative sum down one index (with the surface layer reset to zero) so that each transport bin appears at the depth of its upper boundary.

*6. Some of the discussion of the water mass transformations is more confusing that it is clarifying (see specific comments below).*

**#Response#:** We appreciate the comment and have revised the text to clarify the specific comments drawn on section 4 and improve the overall description of our results.

**Response to RC2's minor comments**

**Section 1**

**L. 33- "at approximately 1000 meters" is misleading, since that is where the streamfunction reaches its maximum, not the northward transport.**
**#Response#:** We thank the reviewer for pointing out the ambiguous sentence. To avoid confusion, we have adjusted our wording accordingly by modifying the sentence to:
*"The maximum AMOC overturning strength occurs within its mid-depth cell, centered around 1000 meters below the ocean surface and between $30°$ to $65°N$."*

**Section 3**

**L. 153-155- What do you mean by "The indices are then further adjusted in density and depth spaces as well in PI and 4xCO2 to capture only the AMOC strength of the upper cell?" Shouldn't you have a generalizable metric that doesn't require manual adjustment in a different climate?**
**#Response#:** With this sentence, we mean that we isolate the upper cell (confined, in PI, between $40 - 60°N$ and $36.20\,\mathrm{kg\,m^{-3}}$ to $36.90\,\mathrm{kg\,m^{-3}}$) according to their minimum and maximum density (depth) during simulation runtime to avoid including any spurious or shallow/tropical/GIN seas recirculation within our derived indices. Our cell-by-cell approach, contrary to a more generalized metric, clearly separates the upper and lower cells under their different timescales of response, including the potential shoaling and poleward shift in these cells and a strengthening of the subtropical cell under warmer climates.

Upon careful analysis of the time evolution of the upper cell against vertical coordinates we defined the AMOC indices in a way that strictly captures the maximum overturning of the upper cell, that relates solely to the Labrador Seawater formation in the Labrador and Irminger Seas. Figure 2 clearly illustrates that not defining the upper and lower limits would in simulation in $4xCO_2$ capture streamfunction values at lighter density classes than those of the upper cell.

We observe that AMOC indices defined in different studies vary, like using the maximum streamfunction at depth excluding the first 500 meters, or just considering a specific latitude for the AMOC indices like the maximum streamfunction between $40 - 60°N$ (Sidorenko et al., 2020a), or at $26°N$ and other fixed latitudes (Frajka-Williams et al., 2019), for example. All of these metrics were defined according to the scope of the

specific study, and we have similarly tailored our analysis towards the scope of our study.

[Figure]

Figure 2: Density and depth of maximum AMOC at $30 - 65°$N with (right) and without (left) upper and lower boundaries.

**L. 168- Okay, but why doesn't this also apply to 4xCO2? Do isopycnals become more titled with climate change?**
**Response#: One might expect that enhanced tilt of isopycnals under $4xCO_2$ would cause the different magnitude and variability in $\rho$- and z-AMOC at $26°$N, but Figure 3 shows virtually identical $\sigma_2$ contours in the subtropics for PI and $4xCO_2$.**

[Figure]

Figure 3: Potential density in $\mathrm{kg\,m^{-3}}$ in the last 50 years of simulation for PI (left panel), $4xCO_2$ (middle panel), and the anomaly between PI and $4xCO_2$ experiments (right panel).

Instead, the divergence between $\rho$- and z-AMOC in $4xCO_2$ arises from a change in interior transformations (Fig 4): whereas in PI both surface and interior transformations act towards denser water mass formations, in $4xCO_2$ there are competing signals of interior transformation towards lighter waters below transformations towards denser waters within the AMOC upper cell ($\sigma_2 = 35\,\mathrm{kg\,m^{-3}}$ to $36.4\,\mathrm{kg\,m^{-3}}$). Because the $\rho$-AMOC is the sum of surface and interior transformations, this net lightening weakens the return flow at $26°$N, driving $\rho$-AMOC variability exhibited in Figure 2 and preventing it from exhibiting the same recovery behavior as

z-AMOC. These dynamics highlight the importance of diagnosing AMOC in depth space, even when one focuses only on the AMOC at 26°N.

[Figure]

Figure 4: Time evolution of interior transformations for PI (left) and $4xCO_2$ (right).

**Section 4**

**L. 258- What do you mean by "interior mixing alone becomes insufficient to sustain deep convection"? Why should we think about interior mixing sustaining deep convection in the first place? Isn't part of the deep mixing is the model \*caused\* by convection, i.e. unstable density profile triggers some kind of deep convection mixing scheme?**
**Response#: We apologise for the confusion caused by our terminology. The correct term is interior transformations, not interior mixing. The usage of the terms interior mixing, interior transformations, and related processes has been reassessed and section 4 has been re-written towards improving the clarity of our results and interpretations.**

**L. 262- What do you mean by "surface transformations trigger interior mixing"? Is this deep convection?**
**Response#: We again apologize for the confusion regarding terminology. As explained in the comment above and in Figure 6 of the manuscript, the surface transformations refer to water mass transformations occurring within the mixed layer that occur due to surface buoyancy fluxes that destabilize the water column and that promote mixing/deep convection. We have re-written section 4 to clarify the terminology, interior mixing vs. interior transformations.**

**Figure 4 - I think you need to expand on this either in the text or the caption to explain to readers how to read these plots, i.e. they are integrated from the North southwards. A meridional derivative in these quantities corresponds to diapycnal transformation whereas it**

**being constant means there is no transformation.**
**#Response#:** We apologise for the lack of clear description of Figure 4. As mentioned in our answer to item 6 of the major comments' section, we have extensively re-written Section 4 to clarify terminology and to guide the reader regarding the interpretation of our results.

**L. 255-257 and Figure 4d- Are you saying that NADW is lighter than AABW? What is going on in the deep density layers? Is this Mediterranean overflow water that is mixing up at high latitudes? I don't really understand how to think about this.**
**#Response#:** We apologise for the confusion in regarding Figure 4d. This figure illustrates the component of the overturning circulation driven by interior transformations, as detailed in lines 248 and 249 of the original manuscript. Consequently, the panels in Figure 4 do not track specific water masses, but rather depict the mechanisms driving the basin-scale overturning through water mass transformations occurring either within the surface mixed layer or in the ocean interior. As indicated in lines 255-257, the Antarctic Bottom Water (AABW) in $4xCO_2$, depicted in Figure 1, occupies lighter density classes with respect to PI due to interior transformations toward lower densities. It should be noted that AABW, as shown in Figure 1, is denser than North Atlantic Deep Water (NADW). As addressed in our response to item 6 of the major comments section, Section 4 has been extensively revised to enhance clarity and facilitate reader comprehension of these figures.

**Section 5**

**L. 307- Be careful here, most of the energy that actually powers the AMOC circulation is mechanically input by Southern Ocean winds or interior turbulent mixing (Wunsch and Ferrari 2004).**
**#Response#:** We appreciate the comment and have revised the sentence to avoid implying direct causality between AMOC strength and diapycnal transformations:
*"Additionally, surface and interior water mass transformations play a crucial role in setting AMOC strength and variability. Therefore, accounting for these transformations is essential to advance our understanding of overturning regimes across various scales."*

**It should be mentioned somewhere that what you call "interior mixing transformation" includes both parameterized physical mixing and spurious numerical mixing (and other residual errors).**
**#Response#:** We thank the reviewer for raising this point. As mentioned before, the correct term that we should use in our manuscript is interior transformations, rather than interior mixing. The usage of the terms interior mixing, interior transformations, and related processes has been reassessed, and Section 4 has been re-written towards improving the clarity of our results. We once more thank the reviewer for drawing our attention to the misapplication of the terms in our manuscript.

**References**

Baker, J. A., Bell, M. J., Jackson, L. C., Vallis, G. K., Watson, A. J., and Wood, R. A.: Continued Atlantic overturning circulation even under climate extremes, Nature, 638, 987–994, https://doi.org/10.1038/s41586-024-08544-0, 2025.

Frajka-Williams, E., Ansorge, I. J., Baehr, J., Bryden, H. L., Chidichimo, M. P., Cunningham, S. A., Danabasoglu, G., Dong, S., Donohue, K. A., Elipot, S., Heimbach, P., Holliday, N. P., Hummels, R., Jackson, L. C., Karstensen, J., Lankhorst, M., Le Bras, I. A., Lozier, M. S., McDonagh, E. L., Meinen, C. S., Mercier, H., Moat, B. I., Perez, R. C., Piecuch, C. G., Rhein, M., Srokosz, M. A., Trenberth, K. E., Bacon, S., Forget, G., Goni, G., Kieke, D., Koelling, J., Lamont, T., McCarthy, G. D., Mertens, C., Send, U., Smeed, D. A., Speich, S., van den Berg, M., Volkov, D., and Wilson, C.: Atlantic Meridional Overturning Circulation: Observed Transport and Variability, Frontiers in Marine Science, 6, 260, https://doi.org/10.3389/fmars.2019.00260, 2019.

Fu, Y., Lozier, M. S., Biló, T. C., Bower, A. S., Cunningham, S. A., Cyr, F., de Jong, M. F., deYoung, B., Drysdale, L., Fraser, N., Fried, N., Furey, H. H., Han, G., Handmann, P., Holliday, N. P., Holte, J., Inall, M. E., Johns, W. E., Jones, S., Karstensen, J., Li, F., Pacini, A., Pickart, R. S., Rayner, D., Straneo, F., and Yashayaev, I.: Seasonality of the Meridional Overturning Circulation in the subpolar North Atlantic, Communications Earth & Environment, 4, 1–13, https://doi.org/10.1038/s43247-023-00848-9, 2023.

Griffies, S. M., Danabasoglu, G., Durack, P. J., Adcroft, A. J., Balaji, V., Böning, C. W., Chassignet, E. P., Curchitser, E., Deshayes, J., Drange, H., Fox-Kemper, B., Gleckler, P. J., Gregory, J. M., Haak, H., Hallberg, R. W., Heimbach, P., Hewitt, H. T., Holland, D. M., Ilyina, T., Jungclaus, J. H., Komuro, Y., Krasting, J. P., Large, W. G., Marsland, S. J., Masina, S., McDougall, T. J., Nurser, A. J. G., Orr, J. C., Pirani, A., Qiao, F., Stouffer, R. J., Taylor, K. E., Treguier, A. M., Tsujino, H., Uotila, P., Valdivieso, M., Wang, Q., Winton, M., and Yeager, S. G.: OMIP Contribution to CMIP6: Experimental and Diagnostic Protocol for the Physical Component of the Ocean Model Intercomparison Project, Geoscientific Model Development, 9, 3231–3296, https://doi.org/10.5194/gmd-9-3231-2016, 2016.

Jackson, L. C. and Petit, T.: North Atlantic Overturning and Water Mass Transformation in CMIP6 Models, Climate Dynamics, 60, 2871–2891, https://doi.org/10.1007/s00382-022-06448-1, 2023.

Megann, A.: Quantifying Numerical Mixing in a Tidally Forced Global Eddy-Permitting Ocean Model, Ocean Modelling, 188, 102 329, https://doi.org/10.1016/j.ocemod.2024.102329, 2024.

Scholz, P., Sidorenko, D., Gurses, O., Danilov, S., Koldunov, N., Wang, Q., Sein, D., Smolentseva, M., Rakowsky, N., and Jung, T.: Assessment of the Finite-volumE Sea ice-Ocean Model (FESOM2.0) – Part 1: Description of selected key model elements and comparison to its predecessor version, Geoscientific Model Development, 12, 4875–4899, https://doi.org/10.5194/gmd-12-4875-2019, 2019.

Sidorenko, D., Danilov, S., Fofonova, V., Cabos, W., Koldunov, N., Scholz, P., Sein, D. V., and Wang, Q.: AMOC, Water Mass Transformations, and Their Responses to Changing Resolution in the Finite-VolumE Sea Ice-Ocean Model, Journal of Advances in Modeling Earth Systems, 12, e2020MS002 317, https://doi.org/10.1029/2020MS002317, 2020a.

Sidorenko, D., Danilov, S., Koldunov, N., Scholz, P., and Wang, Q.: Simple algorithms to compute meridional

overturning and barotropic streamfunctions on unstructured meshes, Geoscientific Model Development, 13, 3337–3345, https://doi.org/10.5194/gmd-13-3337-2020, 2020b.

van Westen, R. M., Kliphuis, M., and Dikjstra, H. A.: Collapse of the Atlantic Meridional Overturning Circulation in a Strongly Eddying Ocean-Only Model, Geophysical Research Letters, 62, e2024GL114 532, https://doi.org/10.1029/2024GL114532, 2025.

---

## Author Comment (AC3)

**Author response for egusphere-2025-2326-EC1**

Fernanda DI Alzira Oliveira Matos, Dmitry Sidorenko, Xiaoxu Shi, Lars Ackermann,
Janini Pereira, Gerrit Lohmann, Christian Stepanek

August 8, 2025

**Letter to the Editor (wrt. EC1)**

Dear Editor,

We are extremely grateful to the two reviewers who raised valid points regarding our submission. We have addressed these comments and have reflected the suggestions in a revised version of our manuscript. Furthermore, we very much appreciate additional comments and suggestions that were raised in EC1. Thanks to the comments by the reviewers and by you we feel that the relevance of our findings will be presented in a more concise and clearer manner. We would like to point out, however, that not all specific points are individually addressed within this response. The manuscript has undergone substantial revision to provide an improved overview of our methodology, a clearer articulation of our motivation, more descriptive results, and a strengthened perspective on the significance of our contribution to the scientific community upon publication of our manuscript. We consequently do not address comments that do not apply anymore as a result of the substantial changes.

Sincerely,

Fernanda D. A. O. Matos

(On behalf of the authors)

**Response to EC1's major comments**

**The article content reads back-to-front to me: having read through it, what I was expecting to see at the front is nested in the middle and the back. The slight result is that (to me at least) it is not entirely clear what the scientific question or motivation is as written relating to the two AMOC diagnostics. In this sense I guess I am biased because I roughly know what the article would be about because I use $\rho$-AMOC quite often (although I don't claim to be in the "serious people" group mentioned by Henri), but the framing is not entirely helpful for**

the more general audience in my opinion.

**#Response#:** We acknowledge that we should have been more clear on our scientific motivation, as well as more careful with structuring our manuscript. We have re-written multiple parts of the manuscript towards successfully transmitting the relevance of our study to the scientific community, and furthermore to address all concerns raised by the reviewers and by you.

**The article currently reads like "we did this and we got this", when it could read more "we think this so we did this and we got this", because some of the content to demonstrate the "we think this" part is actually in the middle/end of the article. The fix is then relatively easy: copy/move/anticipate some of the relevant text discussion up to abstract, introduction and/or section 2. This would help re-balance the article, because section 1 could do with a stronger or more concrete problem statement/ hypothesis, and section 2 as the theoretical/ scientific foundation stone for the article is also a bit short.**

**#Response#:** We thank the editor for this comment and would like to add that this study was motivated by our curiosity on the value of diagnosing $\rho$-AMOC for warmer climates. While existing literature provides substantial insight into $\rho$-AMOC through observational and modelling studies, a gap exists concerning its applicability in the context of abrupt climate change. Our hypothesis is that a transition from z-AMOC to $\rho$-AMOC becomes crucial in warmer climate even where nowadays it is deemed irrelevant to use density space, like in the subtropical North Atlantic, where most studies show that $\rho$- and z-AMOC are quite similar. Our findings definitely corroborate our hypothesis, as the $\rho$-AMOC at 26°N diverges both in magnitude and variability from z-AMOC in our $4xCO_2$ simulation. We also acknowledge that the initial manuscript presentation may have obscured the rationale and significance of this research as well as our motivation. Consequently, the manuscript has been substantially revised to enhance clarity regarding our motivation, the pertinence of results presented, and the rigor of the applied methodology.

**I am also of the strong opinion the authors need to stress that z-AMOC and $\rho$-AMOC are different "diagnostics" and not "AMOCs": there is the model AMOC somehow nested in the diagnosed variables, but there are different representations of it. (cf. a "vector" is the mathematical object, but there are different "representations" of it depending on the choice of basis, and some representations are more useful than others depending on the context.)**

**Following on from that then**

- **The model AMOC is exposed to the same drivers, but this driving is represented differently in the different diagnostics.**

- **The fact they have different distributions and/or magnitudes are not surprising, since they are different diagnostics and measure different things.**

**The article text needs a change of tone and some content to reflect that. The results are fine, but it is a little oversold at the moment to me at least, although I think the referees agree. See "minor comments" of where I think text can be changed/ moved/ copied/ anticipated.**

**#Response#:** We appreciate the insights given through these comments. We have revised the manuscript to clarify that we compare different diagnostics of the same circulation, and we now furthermore argue for the relevance of each diagnostics depending on the scope of one's study.

**As mentioned by the other referee, most MOC calculations use meridional velocity v, so why is w used, particularly when it can be noisy and contribute to uncertainties? Please comment accordingly.**

**#Response#:** For reference, we duplicate here our answer to RC2:

"We appreciate the reviewer's insightful comment, and want to emphasize the robust equivalence of diagnosing MOC using vertical velocities in comparison to the more conventional usage of meridional (horizontal) velocities. Therefore, we expand below on three main aspects:

1. **Equivalence of the two approaches.** Using vertical velocities to compute MOC offers a computationally sound and completely equivalent alternative to integrating binned horizontal velocities. Such equivalence is demonstrated in Sidorenko et al. (2020a,b), wherein the authors explained that binning of horizontal divergence into density classes is done using instantaneous isopycnals, enabling diapycnal velocity calculation after removing the mean drift of isopycnals. The latter is negligibly small in our simulations. Because binning is done with respect to instantaneous isopycnals our AMOC diagnostics are equivalent to those derived using the horizontal velocities. We furthermore note that the use of vertical velocities, instead of horizontal velocities, is more a necessity than a deliberate choice due to the structure of the FESOM2 (see below).

2. **Vertical coordinate in FESOM2.** We apologize for any confusion: FESOM2 does not use Lagrangian vertical coordinate. We implement the Arbitrary Lagrangian-Eulerian (ALE) scheme in a finite volume sense (see lines 80-82 of the original manuscript and Scholz et al. (2019) for more information).

3. **Concern using horizontal velocities.** While computing $\rho$-AMOC using horizontal velocities is feasible and in principal equivalent to MOC computation with vertical velocities, this approach on an unstructured grid, such as the FESOM2 mesh we employ requires careful and non-trivial "broken-line" integration along control-volume boundaries (Sidorenko et al., 2020b) following the discretization of the continuity equation. Furthermore, doing so is less advantageous as it conceals critical information concerning diapycnal velocities. Additionally, diagnosing MOC in density space using vertical velocities has proven more efficient for online diagnostics on the FESOM2 unstructured mesh (Sidorenko et al., 2020a).

We hope this clarifies both the theoretical equivalence and the practical motivations for our chosen diagnostic. We did not, however, include this discussion in our manuscript, for brevity and because it is not within the scope of our study. In the revised manuscript, we included the equation to diagnose MOC using horizontal velocities in the appendix and include the following text to section 2.2:

'In this study, the AMOC is diagnosed using vertical velocities, as opposed to the conventional application of meridional velocities (equations using meridional velocities are detailed in the Appendix). This

*methodology, while mathematically equivalent, presents notable benefits, especially concerning the unstructured FESOM2 mesh, as it directly incorporates diapycnal velocities, and facilitates more efficient online diagnostics (Sidorenko et al., 2020a)."*

Additionally, as our response to RC2 implies, due to the structure of FESOM2, using vertical velocities to diagnose MOC actually introduces less noise and uncertainties in our results.

**Following on from that and as mentioned by the other referee, how is convection represented in the model? Because if it is explicit then it would manifest as a w, but if it is as an enhanced vertical mixing then would one convert a diffusive flux into an effective velocity, or something else? It is thus not clear what $w_\rho$ actually includes, and is therefore not entirely clear what $\psi_\sigma$ is measuring, which is kind of important since that definitions of those are the scientific foundations of the present article. Please clarify accordingly.**

**#Response#:** For reference, we duplicate here our answer to RC1.

"In FESOM2.5, the K-profile parameterization (KPP; Large et al., 1994) scheme was employed for vertical mixing. Consequently, convection is parameterized through enhanced vertical mixing rather than explicit advection of water parcels. In practice, when convective conditions arise, the model artificially increases the vertical diffusivity (on the order of $0.01\,\mathrm{m^2\,s^{-1}}$) to homogenize the water column. This approach is common in hydrostatic ocean models, which cannot explicitly resolve convective plumes or advective exchange across isopycnals due to the hydrostatic approximation. Fully resolving convective overturning would require a non-hydrostatic model with horizontal grid spacing comparable to the vertical scale ($\sim 1\,\mathrm{m}$), an unattainable resolution for global simulations on current supercomputers.

We added the following text into Section 2.1:

*'In FESOM, the K-Profile Parameterization (KPP; Large et al., 1994) scheme is employed for vertical mixing. Consequently, convection arising from local static instability is parameterized through locally enhanced vertical diffusivity, set to $0.01\,\mathrm{m^2\,s^{-1}}$.'"*

**At this resolution some sort of GM scheme is used presumably, then is w\* included in these calculations (probably in $\psi_z$ because GM is supposed to be adiabatic)? This needs clarifying (if GM is not used then please just say so)**
**#Response#:** We thank the editor for highlighting that we should inform the readers about our scheme for parameterizing mesoscale activity. In this sense, we have included the following sentence in Section 2.1:
*"As the low-resolution mesh employed in this study is not eddy-resolving, mesoscale eddy stirring is included via the Gent–McWilliams (GM) parameterization (Gent and McWilliams, 1990) and implemented according to the explicit eddy-induced stream-function algorithm of Ferrari et al. (2010), as described in Danilov et al. (2017) and evaluated in Scholz et al. (2019) for FESOM2."*

**The maths presentation in text and in some of the figures is inconsistent and needs fixing, see below.**
**#Response#:** Based on the further comments outlined in this review, we have re-written the Mathematical Framework section, and included more details on definition of the mathematical equations.

**Response to EC1's minor comments**

**Section 1**

**line 58: Remove comma**
**#Response#:** We appreciate the correction and have removed the comma.

**Section 2**

**Section 2.1**

**line 79: Weird sentence and probably missing the word "unstructured" (because you can't assume people know about details of FESOM). Reword accordingly, e.g. "The unstructured mesh is such that there are approximately 127,000 mesh nodes at the ocean surface." or similar**
**#Response#:** We acknowledge that the description of FESOM2.5 is brief in our manuscript and have extended the model description session to account for more details on all model components.

**line 84: 89 density bins seem a bit small in terms of numbers, and are the bin sizes uniform? Normally I do about 160 to 200 and above uniformly spaced (a bit less if I have it unevenly spaced), but I don't to do averaging in density space online unless I am using MITgcm. Any comments on the dependence/sensitivity to the choice of bin numbers.**
**#Response#:** We acknowledge this concern and, in AWI-CM3, the number of bins used can indeed introduce small-scale recirculations in the diagnosed MOC (Sidorenko et al., 2020a,b, 2021). However, we based our decision to use 89 uneven density bins on the assessment provided by Sidorenko et al. (2020b), wherein the authors describe the sensitivity of AMOC representation to the choice of density bins. For brevity, and because we do not believe it is under the scope of this manuscript, we do not to explicitly include this discussion in our manuscript. We have, however, provided the scripts to generate our plots under Matos (2025), where one can see that the chosen density bins are unevenly spaced and are defined as follows:

*Density bins: 0.0000, 30.00000, 30.55556, 31.11111, 31.36000, 31.66667, 31.91000, 32.22222, 32.46000, 32.77778, 3.01000, 33.33333, 33.56000, 33.88889, 34.11000, 34.44444, 34.62000, 35.00000, 35.05000, 35.10622, 35.20319, 35.29239, 35.37498, 35.41300, 35.45187, 35.52380, 35.59136, 35.65506, 35.71531, 35.77247, 35.82685, 35.87869, 35.92823, 35.97566, 35.98000, 36.02115, 36.06487, 36.10692, 36.14746, 36.18656, 36.22434, 36.26089, 36.29626, 36.33056, 36.36383, 36.39613, 36.42753, 36.45806, 36.48778, 36.51674, 36.54495, 36.57246, 36.59500, 36.59932, 36.62555, 36.65117, 36.67621, 36.68000, 36.70071, 36.72467, 36.74813, 36.75200, 36.77111, 36.79363, 36.81570, 36.83733, 36.85857, 36.87500, 36.87940, 36.89985, 36.91993, 36.93965, 36.95904, 36.97808, 36.99682, 37.01524, 37.03336, 37.05119, 37.06874, 37.08602, 37.10303, 37.11979, 37.13630, 37.15257, 37.16861, 37.18441, 37.50000, 37.75000, 40.00000*

**Section 2.2**

**line 95 + 96: Probably swap colon for a full stop and start a new paragraph as is done already, or follow on straight away (doesn't really matter)**

**Remove all instances of "Eq. X below:", because this is forward referencing and the arguably that text is redundant anyway (don't really need it)**

**Eq 1 and 2: Need to be clear that these are cumulative integrals in y and full integrals in x. In that sense the integral limits need to be $\int_{West}^{y}$.**

**Eq 1 and 2: Why are these flipped from the usual orientation? Would have expected South to North and West to East (which introduces two minus signs that cancel I suppose).**

**Eq 1 and 2: Because of the unstructured mesh one presumably needs to do something in order to do zonal/meridional integrals, so what is actually done? There is a citation to Sidorenko et al (2020a) but this is not that helpful in that there could also be a brief description of what is actually done, because there is unnecessary ambiguity. (Re-interpolation? If so, nearest nearbour, linear or something else? Evaluation of basis element even though this is finite volume?)**

**Equations: Need punctuation to go after them as they should be regarded part of the sentences. So full stops after the symbols at Eq. 1, 3, 5, and commas in Eq. 2 and 4.**

**line 102 + 110: Remove indentation, this is not a new paragraph (don't give it an extra blank line after end equation**

**Response#: We appreciate the comments on Section 2.2 and have completely re-written this part of the manuscript to apply necessary modifications. We modified equations to match completely with the description from Sidorenko et al. (2020a,b, 2021), and provide consistency in comparison with other studies employing a similar mathematical framework (e.g., Xu et al., 2018; Megann, 2018; Megann et al., 2021).**

**Section 3**

**Sec 3 first paragraph: "Averaging in time" is implied but no mention of time window, although this is in Fig 1 caption. This is not entirely helpful, so should mirror that detail around here in the text.**

**Response#: We agree that the time-averaging window should be stated. As a solution, we have added the following statement after the second sentence of the first paragraph in Section 2.1:**

*Analyses of mean large-scale processes were performed using the final 50 years of each simulation.*

**Fig 1 axis labels and elsewhere: Rather than "kgm-3" it should be "kg m-3" and similar. To do this in LaTeX I guess you would do something like "$kg\ m^{-3}$" (as is done in the text). Degrees symbol is slanted here and is inconsistent with how it is used in text; try $^\circ$ if that isn't already what is used (if it is then I don't know what the problem is).**

**Response#: In our manuscript, we have used the latex package `siunitx` that handles SI units. Therefore, we used `degree` unit embedded in the package, for example, for latitude/longitude, and the**

sequence `kilo gram per cubic meter`, for "$kg\ m^{-3}$". As a solution, we replace the `degree` unit with $°$ for the coordinates. However, for other SI units, we see that there is a space (i.e. kg m$^{-3}$) correctly set when using the siunitx package and decided to keep using the `siunitx` package.

**line 142: The two AMOC measures differ in their "spatial distribution" but the comparison shown in Fig 1 is not evidence to support that, because the vertical co-ordinates are completely different. Either**

- **say they differ in the meridional distribution**

- **remap the z-AMOC into density co-ordinates**

- **remap the $\rho$-AMOC into depth co-ordinates if you have a mean isopycnal depth variable computed**

**#Response#:** We thank the reviewer for calling our attention to this expression. We have replaced "spatial" with "latitudinal" to avoid any confusion. In terms of the third item, upon suggestion from Reviewer 2 (see answer to RC2), the $\rho$-AMOC, remapped onto depth coordinates, that was provided in the appendix in the original manuscript, and is now added to the main text in the revised version.

**line 143 + 144: "Distinct driving mechanisms" make no sense to me, because your model is "driven" by the same thing, while the AMOC diagnostics are just that, diagnostics computed from the model variables. Probably lessen or remove related text. Relates to the above point that it is not clear that the two diagnostics are just that, different measures.**
**#Response#:** We thank the editor for drawing our attention to the fact that this sentence is confusing. We acknowledge that the model is driven by the same set of boundary conditions. The intended meaning was to convey that the divergence between the vertical velocity and diapycnal velocity fields stems from an underrepresentation of these driving forces when considering vertical velocity, as opposed to diapycnal velocity. We have addressed this point in the revised manuscript to clarify this contradiction.

**line 155: The "annual and 15 year means" are out of place / unbalanced if there is no "50 year" mean mentioned when talking about the AMOC at the beginning of the section.**
**#Response#:** We apologise for the confusion. As introduced in line 151, the AMOC indices represent the temporal evolution of the AMOC across the 200 simulated years. These indices are graphically depicted in Figure 2, wherein annual means are rendered as thin lines and 15-year rolling means are superimposed as thick lines. To ensure clarity, the text from lines 149-159 has been revised and modified to:

*"Figures 1b and d highlight two major consequences of quadrupling atmospheric $CO_2$ concentrations to the AMOC in both frameworks: the weakening and shoaling of the upper cell (e.g NADW), and the northward-upward expansion of the abyssal cell (e.g. AABW). To assess the consistency of these phenomena across our 200-year integration period, we define two AMOC indices in both density and depth spaces, derived from*

*the streamfunction of each model year: (1) $AMOC_{max}$, denoting the annual maximum overturning between $30 - 65°N$, representing subpolar AMOC; and (2) $AMOC_{26}$, denoting the annual maximum overturning at $26°N$, representing the subtropical AMOC. The upper cell was isolated in both depth and density spaces by implementing the vertical and density boundaries specified in Table 2 for the PI and $4xCO_2$ climates, generating continuous 200-point time series for each index.*

*Figure 2 displays the time series of both indices, with annual means represented by thin lines and 15-year running means superimposed as thick lines to attenuate interannual fluctuations and accentuate multi-decadal variability. A first-degree polynomial trend was fitted and subtracted from the multidecadal time series to facilitate correlation analysis using Pearson's correlation test. Furthermore, the magnitude of AMOC variability was quantified by its standard deviation ($\sigma$). Please note that the previously mentioned 50-year averages relate exclusively to climatological fields (mean state) and not to Figure 2."*

**paragraph beginning line 160: As above, the two AMOC diagnostics are just measuring different things so the differences are not that surprising. They measure different physical effects, so in this case you probably care more about $\rho$-AMOC so just say that.**
**Response**#: We appreciate the comment and these concerns are now addressed in the revised manuscript.**

**line 181 to 187: Would recommend some of this text to be copied/moved up to introduction or section 2 to frame the article more concretely.**
**Response**#: We appreciate the comment and these concerns are now addressed in the revised manuscript.**

**line 199: Don't need the "AMV" acronym because it's never used again anyway.**
**Response**#: We appreciate the correction and have removed this term from the text.**

**Section 4**

**line 219 + 220: Analogously worded sentence should be up in introduction and/or section 2.**
**Response**#: We appreciate the comment and these concerns are now addressed in the revised manuscript.**

**Fig 6 caption (but do this also in text): Over what depth/density classes and averaged how? (full depth?)**
**Response**#: Our choices are as follows:**

- PI: the surface transformations and diapycnal velocities are plotted for the density of $36.68\,\mathrm{kg\,m^{-3}}$, whereas the vertical velocity is plotted for the depth of $910\,\mathrm{m}$.

- $4xCO_2$: the surface transformations and diapycnal velocities are plotted for the density of $35.87\,\mathrm{kg\,m^{-3}}$, whereas the vertical velocity is plotted for the depth of $790\,\mathrm{m}$.

We also provide this information in lines 264-265, where we refer to the table that contains these levels. As for the average, we have added in Section 2.1 that all mean-state plots were obtained through the annual mean average of the last 50 simulation years. We hope that this addition, together with information already provided, suffices for the understanding of the depth/density classes to which the figures are referenced.

**line 263: Either "w$_\rho$" is meant, or need to state why "$w_p$" is different to "w$_\rho$"**
**Response#: We thank the editor for bringing this mistake to our attention. $w_\rho$ is meant.**

**line 280: The model AMOC weakens by the same mechanism presumably but these are projected differently onto the different AMOC diagnostics. Reword accordingly.**
**Response#: Please see answer to comment on lines 143-144. We appreciate the comment and these concerns are now addressed in the revised manuscript.**

**Section 5**

**line 300 to 302: Analogously re-worded sentence should be up in introduction and/or section 2 to anticipate this sentence coming up here.**
**Response#: We appreciate the comment and these concerns are now addressed in the revised manuscript.**

**line 338 to the end: Analogously re-worded paragraph should be up in introduction and/or section 2.**
**Response#: We appreciate the comment and these concerns are now addressed in the revised manuscript.**

**References**

Danilov, S., Sidorenko, D., Wang, Q., and Jung, T.: The Finite-volumE Sea ice–Ocean Model (FESOM2), Geoscientific Model Development, 10, 765–789, https://doi.org/10.5194/gmd-10-765-2017, 2017.

Ferrari, R., Griffies, S. M., Nurser, A. J. G., and Vallis, G. K.: A boundary-value problem for the parameterized mesoscale eddy transport, Ocean Modelling, 32, 143–156, https://doi.org/10.1016/j.ocemod.2010.01.004, the magic of modelling: A special volume commemorating the contributions of Peter D. Killworth – Part 2, 2010.

Gent, P. R. and McWilliams, J. C.: Isopycnal Mixing in Ocean Circulation Models, Journal of Physical Oceanography, 20, 150–155, https://doi.org/10.1175/1520-0485(1990)020¡0150:IMIOCM¿2.0.CO;2, 1990.

Large, W. G., McWilliams, J. C., and Doney, S. C.: Oceanic vertical mixing: A review and a model with a nonlocal boundary layer parameterization, Reviews of Geophysics, 32, 363–403, https://doi.org/https://doi.org/10.1029/94RG01872, 1994.

Matos, F. D. A. O.: Diagnosing the Atlantic Meridional Overturning Circulation in density space is critical under abrupt climate change - Scripts, https://doi.org/10.5281/zenodo.15050831, 2025.

Megann, A.: Estimating the numerical diapycnal mixing in an eddy-permitting ocean model, Ocean Modelling, 121, 19–33, https://doi.org/10.1016/j.ocemod.2017.11.001, 2018.

Megann, A., Blaker, A., Josey, S., New, A., and Sinha, B.: Mechanisms for Late 20th and Early 21st Century Decadal AMOC Variability, Journal of Geophysical Research: Oceans, 126, e2021JC017865, https://doi.org/10.1029/2021JC017865, 2021.

Scholz, P., Sidorenko, D., Gurses, O., Danilov, S., Koldunov, N., Wang, Q., Sein, D., Smolentseva, M., Rakowsky, N., and Jung, T.: Assessment of the Finite-volumE Sea ice-Ocean Model (FESOM2.0) – Part 1: Description of selected key model elements and comparison to its predecessor version, Geoscientific Model Development, 12, 4875–4899, https://doi.org/10.5194/gmd-12-4875-2019, 2019.

Sidorenko, D., Danilov, S., Fofonova, V., Cabos, W., Koldunov, N., Scholz, P., Sein, D. V., and Wang, Q.: AMOC, Water Mass Transformations, and Their Responses to Changing Resolution in the Finite-VolumE Sea Ice-Ocean Model, Journal of Advances in Modeling Earth Systems, 12, e2020MS002317, https://doi.org/10.1029/2020MS002317, 2020a.

Sidorenko, D., Danilov, S., Koldunov, N., Scholz, P., and Wang, Q.: Simple algorithms to compute meridional overturning and barotropic streamfunctions on unstructured meshes, Geoscientific Model Development, 13, 3337–3345, https://doi.org/10.5194/gmd-13-3337-2020, 2020b.

Sidorenko, D., Danilov, S., Streffing, J., Fofonova, V., Goessling, H. F., Scholz, P., Wang, Q., Androsov, A., Cabos, W., Juricke, S., Koldunov, N., Rackow, T., Sein, D. V., and Jung, T.: AMOC Variability and Watermass Transformations in the AWI Climate Model, Journal of Advances in Modeling Earth Systems, 13, e2021MS002582, https://doi.org/10.1029/2021MS002582, 2021.

Xu, X., Rhines, P. B., and Chassignet, E. P.: On Mapping the Diapycnal Water Mass Transformation of the Upper North Atlantic Ocean, Journal of Physical Oceanography, 48, 2233–2258, https://doi.org/10.1175/JPO-D-17-0223.1, 2018.

---

## Referee Report (RR1)

After reading the revised manuscript and the author's response to my previous concerns that clarify their methodology, I have two major concerns regarding the robustness of the methodology:

- 1) As explained in the response to reviewers, the author's method of diagnosing the density-based overturning streamfunction is only equivalent with the more conventional meridional transport-based method of diagnosing the density-based overturning if isopycnal drifts are taken into account. However, the authors provide zero evidence that the drift is negligible, either in the revised manuscript, in their response to my concern, or in the cited Sidorenko (2020) paper which establishes the methodology. Based on my experience performing similar calculations, I can confirm that mean isopycnal drifts are generally negligible contributions to piControl water mass budgets on multi-decadal timescales; in strongly-forced runs, however, I have found that the drift is a leading-order term in the water mass budget! For a manuscript focusing on the response of the density-based overturning to strong forcing, it is unacceptable to leave this key supporting evidence as "(not shown)".
- 2) Figure 2 appears to reveal a fatal flaw in the methodology. The fact that the density of the maximum \rho-AMOC strength oscillations between about ~31 kg/m^3 and 36.4 kg/m^3 suggests to me that this metric is not robustly picking up the AMOC cell. While the authors seem to recognize this and have thus constrained the range of valid densities (as shown in their Figure 2b reproduced below) to a fixed range. [Aside: The main text states that the upper boundary is 35, whereas this figure shows it to be 34. Which is the correct one?] However, the maximum density still jumps around, just now within the artificially constrained range. By visual inspection, it seems to me that these spurious multi-decadal jumps in the density of the rho-AMOC map onto the multi-decadal oscillations in the strength of the rho-AMOC streamfunction, which the authors argue is a key result of their paper, which is supposedly hidden by the z-AMOC streamfunction.

I am concerned that these issues could constitute fatal flaws that compromise the key results of the manuscript. I would be willing to conduct a more thorough review after the authors have more substantially addressed these concerns.

---

## Author Response (AR2)

**Author response for egusphere-2025-2326**

Fernanda DI Alzira Oliveira Matos, Dmitry Sidorenko, Xiaoxu Shi, Lars Ackermann, Janini Pereira, Gerrit Lohmann, Christian Stepanek

September 23, 2025

**Letter to the Editor and to the Reviewers**

Dear Editor, and Reviewer 2,

We deeply appreciate the points raised by both of you to the second version of our manuscript. We have addressed these comments and have reflected the suggestions in a revised version of our manuscript.

Sincerely,

Fernanda D. A. O. Matos

(On behalf of the authors)

**Response to the Editor**

abstract: probably "...maximum is substantially stronger than THAT OF THE z-AMOC, ... #Response#: We appreciate the comment and have corrected the sentence accordingly.

abstract: comma, before "which", "...mass transformations, which are concealed... #Response#: We appreciate the comment and have corrected the sentence accordingly.

introduction: last sentence of paragraph 5 beginning "Moreover, deriving the overturning circulation in density space..." should be moved up. This is known, and if anything should be the motivating factor for the rest of the paragraph.

**Response#: We thank the reviewer for this helpful suggestion. We agree that the broader motivation should appear earlier. We have therefore repositioned the sentence from the end of the paragraph to an earlier position. It now follows the explanation that zonal averaging in depth space conceals the horizontal separation between the AMOC limbs in the subpolar North Atlantic and precedes the concluding statement.**

This placement highlights the general applicability beyond the North Atlantic and present climate while preserving the logical sequence of the argument from the mechanism in the subpolar North Atlantic, broader relevance, conclusion, and observed community uptake. The paragraph is now written as:

"The choice of coordinate system becomes particularly relevant when comparing the overturning estimates in the SPNA with those in the subtropical North Atlantic (STNA). In the STNA, the strong stratification and relatively flat isopycnals allow the southward limb to flow directly beneath the northward upper limb, resulting in similar estimates of magnitude and variability of  $\rho$ - and z-AMOC (Moat et al., 2025). Conversely, in the SPNA, sloped isopycnals induce horizontal separation between the upper and lower limbs, resulting in divergence between  $\rho$ - and z-AMOC in terms of both strength and variability (Foukal and Chafik, 2024). This divergence arises because zonal averaging in depth space conceals this horizontal separation, which in turn compromises the accuracy of AMOC estimates under this representation, particularly in the SPNA. Moreover, deriving the overturning circulation in density space instead of depth space is advantageous beyond studies that focus mostly on the North Atlantic and on current climate change, as constant-depth averaging can lead to spurious features such as the Deacon cell in the Southern Ocean (Döös and Webb, 1994; Stevens and Ivchenko, 1997; Speer et al., 2000) and has been linked to discrepancies between modeled and observed AMOC variability across timescales (Liu et al., 2017). Thus, diagnosing AMOC in density space yields a more continuous and physically consistent representation of the AMOC and its underlying mechanisms (Megann, 2018; Xu et al., 2018; Sidorenko et al., 2020a, 2021; Megann et al., 2021; Foukal and Chafik, 2024)"

Introduction: paragraph 6 ordering of reasoning is back to front. The computation cost is high so people don't do it. As written it somewhat implies people don't do it for no particularly good reason when they should, but then justifies it by the computation cost, which is internally inconsistent.

**Response#: We thank the editor for this comment. We have revised the paragraph to the following:**

"Although diagnosing AMOC in density space provides clear advantages, the establishment of  $\rho$ -AMOC as the standard diagnostic is still hindered by the research community's long-standing familiarity with z-AMOC. built over decades of studies employing the latter definition, with some articles providing a supplementary figure of  $\rho$ -AMOC remapped onto depth coordinates to facilitate comparisons between depth and density space AMOC representations (e.g. Xu et al., 2018; Tesdal et al., 2023; Foukal and Chafik, 2024). An additional caveat includes the higher computational cost associated with diagnosing  $\rho$ -AMOC (Sidorenko et al., 2021), which can discourage its implementation in studies that require long integration periods or high-resolution output. Furthermore, while the streamfunction in density space has been requested as output in CMIP6 (Griffies et al., 2016), it was not provided consistently for all experiments by all participating modelling centers (Baker et al., 2025; Jackson and Petit, 2023), which limits model intercomparison. While these barriers remain, the scientific gain through diagnosing  $\rho$ -AMOC outweighs these challenges, driving its increasing recognition in recent decades, with a strong momentum towards diagnosing AMOC in density space either in modelling studies or observational arrays at various latitudes (e.g. Frajka-Williams et al., 2023; Jackson and Petit, 2023; Fu et al., 2023; van Westen et al., 2025). In particular, observational arrays such as the OSNAP (Overturning in the Subpolar North Atlantic Program; Lozier et al., 2017) and RAPID-MOCHA (RAPID Climate Change - Meridional Overturning Circulation and Heatflux Array; McCarthy et al., 2015),

already provide  $\rho$ -AMOC output and have, since their launch, changed our view on overturning in the subpolar and subtropical North Atlantic (Lozier et al., 2019; Moat et al., 2025; Frajka-Williams et al., 2023). Consequently, this dichotomy between the studies employing  $\rho$ - and/or z-AMOC frameworks introduces increased uncertainty regarding the occurrence and timing of a substantial AMOC weakening under a warming climate. At the inception of CMIP7, where both  $\rho$ - and z-AMOC are requested (Fox-Kemper et al., 2025, in review), we see a timely opportunity to advertise the more widespread adoption of  $\rho$ -AMOC, at least as an additional, if not even the main, overturning diagnostic."

Same paragraph: grammar, "...and have, since their LAUNCH, changed..."

**Response#: We appreciate the comment and have corrected the sentence accordingly.**

Eq (1): still unclear whether the w\* resulting from GM scheme (which is active) is included here. (It shouldn't be in Eq. (2) if that is a diapycnal velocity)

**Response#: We thank the reviewer for pointing this out. In Eq.(1) we use the vertical velocity that includes the GM bolus contribution  $w^*$ . However, in Eq.(2),  $w_{\rho}$  as the GM transport is adiabatic, the  $w_{\rho}^*$  is not included in  $w_{\rho}$ . We have added clarifying sentence immediately below Eq.1.**

"where w denotes the vertical velocity that includes the GM bolus component  $w^*$ ."

Below Eq (4): "...model drift was calculated and found TO BE negligible." #Response#: We appreciate the comment and have corrected the sentence accordingly.

Paragraph above Sec 3: they are only mathematically equivalent if the flow is incompressible, but there is no mention of that property being satisfied by the present model explicitly (if it is hydrostatic I guess it's implied, but only implicitly)

**Response#: We agree that both MOC formulations are equivalent under incompressibility. FESOM2 solves the hydrostatic primitive equations under the Boussinesq (incompressible) approximation. Therefore the two formulations are formally equivalent in this model (up to negligible discretization error; see Banerjee et al. (2024)). We have modified the following paragraph to explicitly inform the readers about how FESOM2 solves primitive equations:**

**In the mathematical framework section (paragraph above section 3)**

"Please note that, in our study, the AMOC is computed using vertical velocity, w, rather than the conventional approach based on meridional velocity (v; see Table A1 for the equations). We make this choice because the hydrostatic Boussinesq formulation used by FESOM2 implies incompressibility (Banerjee et al., 2024), which makes both methodologies mathematically equivalent (see Sidorenko et al., 2020b, for a detailed comparison) and because not using meridional velocity is advantageous with the spatial discretization used by FESOM2. Using the meridional velocity would require integration along the boundaries of the control volumes, which is less convenient for arbitrary unstructured meshes, and it would also neglect important information about diapycnal velocities. In contrast, the vertical velocity approach naturally yields the AMOC in density space, reduces noise introduced by the beforementioned integration along the boundaries

of the control volumes, and enables more efficient online diagnostics in FESOM2 (Sidorenko et al., 2020a, 2021). Additionally, since model drift is negligible during model runtime and at the density range of the upper cell (Figure A2), surface and interior transformations constitute the only  $\rho$ -AMOC components in our simulations"

Sentence near Table 2: should just be "...oscillatING around 5 Sc, suggesting..." #Response#: We appreciate the comment and have corrected the sentence accordingly.

if the authors deem it appropriate please acknowledge all those who provided comments accordingly.

**Response#: We thank the editor for drawing our attention to this, we have updated the last two sentences of the Acknowledgements section to:**

"We also acknowledge the Nippon Foundation and the Partnership for the Observation of the Global Ocean (POGO) for support under the NF-POGO Centre of Excellence (NF-POGO CofE) programme. Finally, we thank Dr. Paul Gierz and Dr. Sergey Danilov for their valuable guidance during the preparation and revision of this article. We are extremely grateful to the editor, Dr. Julian Mak, the reviewer, Dr. Henri Drake, and one more anonymous reviewer for their constructive feedback and comments that have greatly increased the quality and comprehensibility of our manuscript."

**Response to Reviewer 2**

As explained in the response to reviewers, the author's method of diagnosing the density-based overturning streamfunction is only equivalent with the more conventional meridional transport-based method of diagnosing the density-based overturning if isopycnal drifts are taken into account. However, the authors provide zero evidence that the drift is negligible, either in the revised manuscript, in their response to my concern, or in the cited Sidorenko (2020) paper which establishes the methodology. Based on my experience performing similar calculations, I can confirm that mean isopycnal drifts are generally negligible contributions to piControl water mass budgets on multi-decadal timescales; in strongly-forced runs, however, I have found that the drift is a leading-order term in the water mass budget! For a manuscript focusing on the response of the density-based overturning to strong forcing, it is unacceptable to leave this key supporting evidence as "(not shown)".

**Response#: We thank the reviewer for emphasizing the role of isopycnal drift under strong forcing. We now display the isopycnal volume tendency ("drift term") in figure A2 (Figure 1 here) in the appendix. During model runtime at the latitude of stronger AMOC, Figure 1 reveals that the drift is negligible. Thus, the weakening and multidecadal variability of the upper overturning cell in the 4xCO2 persist and remain consistent as a response to such an abrupt forcing.**

Figure 1: Hovmöller diagrams of  $\rho$ -AMOC (a,c) and model drift (b,d) in density space for PI (a,b) and  $4xCO_2$  (c,d). Red dashed lines denote the upper and lower density bounds of the  $\rho$ -AMOC upper limb, computed from the mean over the last 50 years of each simulation. The  $\rho$ -AMOC maximum latitude for both simulations is displayed in Table 1.

Figure 2 appears to reveal a fatal flaw in the methodology. The fact that the density of the maximum  $\rho$ -AMOC strength oscillations between about  $\sim 31$  kg/m-3 and 36.4 kg/m-3 suggests to me that this metric is not robustly picking up the AMOC cell. While the authors seem to recognize this and have thus constrained the range of valid densities (as shown in their Figure 2b reproduced below) to a fixed range. [Aside: The main text states that the upper boundary is 35, whereas this figure shows it to be 34. Which is the correct one?] However, the maximum density still jumps around, just now within the artificially constrained range. By visual inspection, it seems to me that these spurious multi-decadal jumps in the density of the rho-AMOC map onto the multi-decadal oscillations in the strength of the rho-AMOC streamfunction, which the authors argue is a key result of their paper, which is supposedly hidden by the z-AMOC streamfunction.

**Response#: We thank the reviewer for this constructive suggestion. In the revised manuscript we clarify that the variability captured by our indices reflects physical water—mass transformation processes rather than an artefact of index definition. Motivated by the comment, we extended the analysis to derive an index that considers the full density range and compared it with our original index (Figure 2 in this response). This additional comparison shows that incorporating the full density range does not affect the variability of z- and  $\rho$ -AMOCmax during model runtime, it does only modify the strength of z-AMOCmax (as expected). As the reviewer noted, the 4xCO2 experiment represents a strongly forced climate, distinct from more weakly forced cases like piControl, which therefore alters the AMOC variability. Yet, based on the additional analyses that we show now we can explicitly state that the variability in AMOC in the 4xCO2 experiment is not an artifact of the chosen density classes.**

Figure 2: Comparison between  $AMOC_{max}$  indices. In a) the original index, derived from the maximum AMOC strength between 30 - 65°N. In (b, c, d) the indices derived within the entire density range but with different latitudinal boundaries.

 $AMOC_{max}$  is now defined as the basin maximum of the overturning streamfunction over the full density range north of 50°N. This cutoff lies within the latitude window used in previous studies (e.g.; Cheng et al., 2013; Ackermann et al., 2020; Matos et al., 2020; Sidorenko et al., 2021) and targets the subpolar deep-water formation regions, while excluding subtropical contributions. By taking the maximum over the entire density axis, no upper or lower density bounds need to be prescribed, as evinced in Figure 3:

Figure 3: Density of maximum  $\rho$ -AMOC strength north of a) 35°N, b) 37°N, c) 40°N, d) 50°N

We are extremely grateful to the reviewer to have pointed out that our analysis, as presented in the previous version of the manuscript, where we had not discussed the impact of changes in density classes on our results, may lead to doubts regarding robustness of our inferences. Their comments have greatly improved presentation of our results towards robustness and comprehension of our study. Consequently, we have updated the text and the figure in the revised manuscript accordingly.

**References**

- Ackermann, L., Danek, C., Gierz, P., and Lohmann, G.: AMOC Recovery in a Multicentennial Scenario Using a Coupled Atmosphere-Ocean-Ice Sheet Model, Geophysical Research Letters, 47, e2019GL086810, https://doi.org/10.1029/2019GL086810, 2020.
- Baker, J. A., Bell, M. J., Jackson, L. C., Vallis, G. K., Watson, A. J., and Wood, R. A.: Continued Atlantic overturning circulation even under climate extremes, Nature, 638, 987–994, https://doi.org/10.1038/s41586-024-08544-0, 2025.
- Banerjee, T., Scholz, P., Danilov, S., Klingbeil, K., and Sidorenko, D.: Split-explicit external mode solver in the finite volume sea ice-ocean model FESOM2, Geoscientific Model Development, 17, 7051–7065, https://doi.org/10.5194/gmd-17-7051-2024, 2024.
- Cheng, W., Chiang, J. C. H., and Zhang, D.: Atlantic Meridional Overturning Circulation (AMOC) in CMIP5 Models: RCP and Historical Simulations, Journal of Climate, pp. 7187–7197, https://doi.org/10.1175/JCLI-D-12-00496.1, 2013.
- Döös, K. and Webb, D. J.: The Deacon Cell and the Other Meridional Cells of the Southern Ocean, Journal of Physical Oceanography, 24, 429–442, https://doi.org/10.1175/1520-0485(1994)024\langle0429:TDCATO\rangle2.0. CO;2, 1994.
- Foukal, N. P. and Chafik, L.: Consensus Around a Common Definition of Atlantic Overturning Will Promote Progress, Oceanography, 37, 10–15, https://doi.org/10.5670/oceanog.2024.507, 2024.
- Fox-Kemper, B., DeRepentigny, P., Treguier, A. M., Stepanek, C., O'Rourke, E., Mackallah, C., Meucci, A., Aksenov, Y., Durack, P. J., Feldl, N., Hernaman, V., Heuzé, C., Iovino, D., Madan, G., Marquez, A. L., Massonnet, F., Mecking, J., Samanta, D., Taylor, P. C., Tseng, W.-L., and Vancoppenolle, M.: CMIP7 Data Request: Ocean and Sea Ice Priorities and Opportunities, EGUsphere, 2025, 1–58, https://doi.org/10.5194/egusphere-2025-3083, 2025.
- Frajka-Williams, E., Foukal, N., and Danabasoglu, G.: Should AMOC observations continue: how and why?, Philosophical Transactions of the Royal Society A: Mathematical, Physical and Engineering Sciences, 381, 20220 195, https://doi.org/10.1098/rsta.2022.0195, 2023.
- Fu, Y., Lozier, M. S., Biló, T. C., Bower, A. S., Cunningham, S. A., Cyr, F., de Jong, M. F., deYoung, B., Drysdale, L., Fraser, N., Fried, N., Furey, H. H., Han, G., Handmann, P., Holliday, N. P., Holte, J., Inall, M. E., Johns, W. E., Jones, S., Karstensen, J., Li, F., Pacini, A., Pickart, R. S., Rayner, D., Straneo, F., and Yashayaev, I.: Seasonality of the Meridional Overturning Circulation in the subpolar North Atlantic, Communications Earth & Environment, 4, 1–13, https://doi.org/10.1038/s43247-023-00848-9, 2023.

- Griffies, S. M., Danabasoglu, G., Durack, P. J., Adcroft, A. J., Balaji, V., Böning, C. W., Chassignet, E. P., Curchitser, E., Deshayes, J., Drange, H., Fox-Kemper, B., Gleckler, P. J., Gregory, J. M., Haak, H., Hallberg, R. W., Heimbach, P., Hewitt, H. T., Holland, D. M., Ilyina, T., Jungclaus, J. H., Komuro, Y., Krasting, J. P., Large, W. G., Marsland, S. J., Masina, S., McDougall, T. J., Nurser, A. J. G., Orr, J. C., Pirani, A., Qiao, F., Stouffer, R. J., Taylor, K. E., Treguier, A. M., Tsujino, H., Uotila, P., Valdivieso, M., Wang, Q., Winton, M., and Yeager, S. G.: OMIP Contribution to CMIP6: Experimental and Diagnostic Protocol for the Physical Component of the Ocean Model Intercomparison Project, Geoscientific Model Development, 9, 3231–3296, https://doi.org/10.5194/gmd-9-3231-2016, 2016.
- Jackson, L. C. and Petit, T.: North Atlantic Overturning and Water Mass Transformation in CMIP6 Models, Climate Dynamics, 60, 2871–2891, https://doi.org/10.1007/s00382-022-06448-1, 2023.
- Liu, W., Xie, S.-P., Liu, Z., and Zhu, J.: Overlooked possibility of a collapsed Atlantic Meridional Overturning Circulation in warming climate, Science Advances, 3, e1601666, https://doi.org/10.1126/sciadv.1601666, 2017.
- Lozier, M. S., Bacon, S., Bower, A. S., Cunningham, S. A., de Jong, M. F., de Steur, L., deYoung, B.,
  Fischer, J., Gary, S. F., Greenan, B. J. W., Heimbach, P., Holliday, N. P., Houpert, L., Inall, M. E.,
  Johns, W. E., Johnson, H. L., Karstensen, J., Li, F., Lin, X., Mackay, N., Marshall, D. P., Mercier,
  H., Myers, P. G., Pickart, R. S., Pillar, H. R., Straneo, F., Thierry, V., Weller, R. A., Williams, R. G.,
  Wilson, C., Yang, J., Zhao, J., and Zika, J. D.: Overturning in the Subpolar North Atlantic Program: A
  New International Ocean Observing System, Bulletin of the American Meteorological Society, 98, 737–752,
  https://doi.org/10.1175/BAMS-D-16-0057.1, 2017.
- Lozier, M. S., Li, F., Bacon, S., Bahr, F., Bower, A. S., Cunningham, S. A., de Jong, M. F., de Steur, L., deYoung, B., Fischer, J., Gary, S. F., Greenan, B. J. W., Holliday, N. P., Houk, A., Houpert, L., Inall, M. E., Johns, W. E., Johnson, H. L., Johnson, C., Karstensen, J., Koman, G., Le Bras, I. A., Lin, X., Mackay, N., Marshall, D. P., Mercier, H., Oltmanns, M., Pickart, R. S., Ramsey, A. L., Rayner, D., Straneo, F., Thierry, V., Torres, D. J., Williams, R. G., Wilson, C., Yang, J., Yashayaev, I., and Zhao, J.: A sea change in our view of overturning in the subpolar North Atlantic, Science, 363, 516–521, https://doi.org/10.1126/science.aau6592, 2019.
- Matos, F. D. A. O., Pereira, J., and Dengler, M.: Salinity Biases and the Variability of the Atlantic Meridional Overturning Circulation in GFDL-CM3, Ocean Science Journal, 55, 505–520, https://doi.org/10.1007/s12601-020-0040-8, 2020.
- McCarthy, G. D., Smeed, D. A., Johns, W. E., Frajka-Williams, E., Moat, B. I., Rayner, D., Baringer, M. O., Meinen, C. S., Collins, J., and Bryden, H. L.: Measuring the Atlantic Meridional Overturning Circulation at 26°N, Progress in Oceanography, 130, 91–111, https://doi.org/10.1016/j.pocean.2014.10.006, 2015.
- Megann, A.: Estimating the numerical diapycnal mixing in an eddy-permitting ocean model, Ocean Modelling, 121, 19–33, https://doi.org/10.1016/j.ocemod.2017.11.001, 2018.
- Megann, A., Blaker, A., Josey, S., New, A., and Sinha, B.: Mechanisms for Late 20th and Early 21st Century Decadal AMOC Variability, Journal of Geophysical Research: Oceans, 126, e2021JC017865, https://doi.org/10.1029/2021JC017865, 2021.

- Moat, B., Smeed, D., Rayner, D., Johns, W., Smith, R., Volkov, D., Elipot, S., Petit, T., Kajtar, J., Baringer, M. O., and Collins, J.: Atlantic meridional overturning circulation observed by the RAPID-MOCHA-WBTS (RAPID-Meridional Overturning Circulation and Heatflux Array-Western Boundary Time Series) array at 26N from 2004 to 2023 (v2023.1a), https://doi.org/10.5285/33826d6e-801c-b0a7-e063-7086abc0b9db, 2025.
- Sidorenko, D., Danilov, S., Fofonova, V., Cabos, W., Koldunov, N., Scholz, P., Sein, D. V., and Wang, Q.: AMOC, Water Mass Transformations, and Their Responses to Changing Resolution in the Finite-VolumE Sea Ice-Ocean Model, Journal of Advances in Modeling Earth Systems, 12, e2020MS002317, https://doi.org/10.1029/2020MS002317, 2020a.
- Sidorenko, D., Danilov, S., Koldunov, N., Scholz, P., and Wang, Q.: Simple algorithms to compute meridional overturning and barotropic streamfunctions on unstructured meshes, Geoscientific Model Development, 13, 3337–3345, https://doi.org/10.5194/gmd-13-3337-2020, 2020b.
- Sidorenko, D., Danilov, S., Streffing, J., Fofonova, V., Goessling, H. F., Scholz, P., Wang, Q., Androsov, A., Cabos, W., Juricke, S., Koldunov, N., Rackow, T., Sein, D. V., and Jung, T.: AMOC Variability and Watermass Transformations in the AWI Climate Model, Journal of Advances in Modeling Earth Systems, 13, e2021MS002582, https://doi.org/10.1029/2021MS002582, 2021.
- Speer, K., Rintoul, S. R., and Sloyan, B.: The Diabatic Deacon Cell, Journal of Physical Oceanography, 30, 3212–3222, https://doi.org/10.1175/1520-0485(2000)030(3212:TDDC)2.0.CO;2, 2000.
- Stevens, D. P. and Ivchenko, V. O.: The zonal momentum balance in an eddy-resolving general-circulation model of the southern ocean, Quarterly Journal of the Royal Meteorological Society, 123, 929–951, https://doi.org/10.1002/qj.49712354008, 1997.
- Tesdal, J.-E., MacGilchrist, G. A., Beadling, R. L., Griffies, S. M., Krasting, J. P., and Durack, P. J.: Revisiting interior water mass responses to surface forcing changes and the subsequent effects on overturning in the Southern Ocean, Journal of Geophysical Research: Oceans, 128, e2022JC019105, https://doi.org/10.1029/2022JC019105, 2023.
- van Westen, R. M., Kliphuis, M., and Dikjstra, H. A.: Collapse of the Atlantic Meridional Overturning Circulation in a Strongly Eddying Ocean-Only Model, Geophysical Research Letters, 62, e2024GL114532, https://doi.org/10.1029/2024GL114532, 2025.
- Xu, X., Rhines, P. B., and Chassignet, E. P.: On Mapping the Diapycnal Water Mass Transformation of the Upper North Atlantic Ocean, Journal of Physical Oceanography, 48, 2233–2258, https://doi.org/10.1175/JPO-D-17-0223.1, 2018.

---

## Author Response (AR4)

**Author response for egusphere-2025-2326**

Fernanda DI Alzira Oliveira Matos, Dmitry Sidorenko, Xiaoxu Shi, Lars Ackermann, Janini Pereira, Gerrit Lohmann, Christian Stepanek

October 15, 2025

**Letter to the Editor and to the Reviewers**

Dear Editor, and Reviewer 2,

We deeply appreciate the points raised in this last iteration of the review process of our manuscript. From the ten comments provided by Reviewer 2, we have addressed all those related to textual clarity by rephrasing the sentences to improve readability and consistency with the results. We have also reorganized the items of the mathematical framework to include the model drift component before the interior transformations and have retained Figure A2, as it demonstrates that the model drift is negligible. In addition, we have addressed the concern regarding the clarity of the multidecadal oscillation in the Arctic winter sea-ice time series shown in Figure 3 by adding a superimposed fifteen-year rolling mean, which highlights the low-frequency variability more clearly.

Regarding Figure 1, we decided not to modify the remapped  $\rho$ -AMOC field as suggested by the reviewer. The alternative approach, mentioned by Reviewer 2, was carefully tested during the development of the diagnostic tools used in this study and was found to introduce numerical artifacts and to fail to ensure monotonic vertical coordinates of the density classes. The approach adopted in our analysis instead relies on the cumulative sum of density-class layer thicknesses to obtain the vertical coordinates, which guarantees monotonicity and a physically consistent streamfunction. This implementation was thoroughly validated using the FESOM, ICON, and MITgcm models, and consistently produced the most robust and interpretable results. For these reasons, and to maintain consistency with the established FESOM diagnostics, we have chosen to retain the current version of Figure 1.

Sincerely,

Fernanda D. A. O. Matos

(On behalf of the authors)